# India Indenting Eurasia: A Brief Review and New Data from the Yongping Basin on the SE Tibetan Plateau

Tiannan Yang [1,2,*], Zhen Yan [2], Chuandong Xue [3], Di Xin [2] and Mengmeng Dong [2]

1   Key Laboratory of Deep-Earth Dynamics of Ministry of Natural Resources, Beijing 100037, China
2   Institute of Geology, Chinese Academy of Geological Sciences, Beijing 100037, China; yanzhen@mail.iggcas.ac.cn (Z.Y.); xindi@cags.ac.cn (D.X.); dongmengmengm@163.com (M.D.)
3   Department of Earth Sciences, Kunming University of Sciences and Technology, Kunming 650093, China; xuechuandong@kust.edu.cn
*  Correspondence: yangtn@cags.ac.cn; Tel.: +86-1068999722

**Abstract:** Successive indentations of Eurasia by India have led to the Tibet-Himalaya E–W orthogonal collision belt and the SE Tibetan Plateau N–S oblique collision belt along the frontal and eastern edges of the indenter, respectively. The belts exhibit distinctive lithospheric structures and tectonic evolutions. A comprehensive compilation of available geological and geophysical data reveals two sudden tectonic transitions in the early Eocene and the earliest Miocene, respectively, of the tectonic evolution of the orthogonal belt. Synthesizing geological and geochronological data helps us to suggest a NEE–SWW trending, ~450 km-long, ~250 km-wide magmatic zone in SE Tibet, which separates the oblique collision belt (eastern and SE Tibet) into three segments of distinctive seismic structures including the mantle and crust anisotropies. The newly identified Yongping basin is located in the central part of the magmatic zone. Geochronological and thermochronological data demonstrate that (1) this basin and the magmatic zone started to form at ~48 Ma likely due to NNW–SSE lithosphere stretching according to the spatial coincidence of the concentrated mantle-sourced igneous rocks on the surface with the seismic anomalies at depth; and (2) its fills was shortened in the E–W direction since ~23 Ma. These two dates correspond to the onset of the first and second tectonic transitions of the orthogonal collision belt. As such, both the orthogonal and oblique belts share a single time framework of their tectonic evolution. By synthesizing geological and geophysical data of both collision belts, the indenting process can be divided into three stages separated by two tectonic transitions. Continent–continent collision as a piston took place exclusively during the second stage. During the other two stages, the India lithosphere underthrust beneath Eurasia.

**Keywords:** indenting process; continental collision; lithosphere underthrust; oblique collision belt; tectonic transitions; Eocene; Early Miocene; SE Tibet

## 1. Introduction: A Brief Review of Cenozoic Geology of the Tibetan Plateau

The Tibetan Plateau is an ideal natural laboratory for studies of continental collision dynamics [1,2] and has been a focus of multidisciplinary research for more than a century. The accumulation of geological and geophysical data has revealed the following two traits of the plateau.

(1)     Two collision belts with distinctive lithospheric structures

The Indian continent has indented northwards into Eurasia by at least 2000 km [3,4]. The progressive indenting has led to an orthogonal collision belt along the frontal edge and an oblique collision belt along the eastern edge of the indenter [4]. These belts are known as the Tibet-Himalaya and the Southeastern (SE) Tibetan Plateau [5], respectively, and they are currently separated by the Eastern Himalaya Syntax (EHS) [6].

The Tibet-Himalaya is the "center of the storehouse of excess gravitational potential energy accumulated through crustal thickening" [7]. Balancing the accumulating potential

energy due to the continuous orthogonal collision of India by dissipation along the margins of the plateau has led to a "steady-state" geometric framework of the Tibet-Himalaya, where the styles of deformation have been maintained for at least 20 myr [7]. This model is consistent with the apparent agreement between the current-day surface deformation (e.g., [8,9]) and the longer-term deformation of the lithosphere as recorded by the anisotropy of the crust and mantle [10,11].

By comparison, the oblique collision belt grew and will continue to grow progressively northwards due to the ongoing northwards indenting of India into Eurasia. GPS geodetic studies [8,9,12–14] have revealed the present-day flow of Tibetan crustal material around the EHS in a reference frame fixed to Eurasia and the flow is clockwise around the syntax, including a southwestward motion in western Yunnan, SW China, and in eastern Myanmar. This velocity field is polar in that it can be described approximately by small circles about a pole near the EHS [15,16]. Although the surficial velocity of the plateau (including the SE Tibetan Plateau) varies smoothly and gradually along its strike [8,9], a sharp transition in mantle anisotropy has been identified across a geophysically well-known lithospheric boundary (LB), the Latitude 26° N Line on the SE Tibetan Plateau [13,17]. This NEE–SWW-trending boundary occurs between about latitudes 24° N and 27° N (Figure 1), which extends in the NEE direction for at least 450 km [17]. South of the LB, shear-wave splitting exhibits a uniform pattern of E–W fast directions [13,17,18] that is sub-perpendicular to the current flow of crustal materials. P-wave [19,20] and Rayleigh wave anisotropic tomography [21,22] demonstrate highly spatially variable deformation patterns of the crust and the upper mantle beneath SE Tibet, indicating that the long-term deformation of the oblique collisional belt is apparently inconsistent with the current-day surface deformation as inverted from the GPS observations [9,20,23,24].

Moreover, the seismic structures of the lithosphere beneath SE Tibet do not well match its Cenozoic surficial structures. The latter have long been thought of being characterized by several NW–SE trending lithosphere-scale lateral sliding fault zones (e.g., [25] and references therein), whereas most seismic anomalies [19,22,26] are sub-perpendicular to the so-called lateral sliding faults. This inconsistency suggests a secular change in the longer-term lithosphere deformation of the oblique collision belt. At the present time, however, we know very little about this change (e.g., [27]).

(2)    Three stages of "steady-state" convergence separated by two tectonic transitions

Paleomagnetic measurements from the 90° E ridge of the Indian Ocean [28], in addition to other studies (e.g., [29,30]) have shown that the Indian Plate has shifted with a variable velocity in the reference frame of a fixed Eurasia during the Cretaceous to Cenozoic, which implies variable convergence rates between India and Eurasia. Lee and Lawyer [31] calculated the relative motions of India and Eurasia according to the relative motions of India and an absolute framework, between Eurasia and the same absolute framework, and between the same absolute framework and North America. Molnar et al. [30] and Molnar and Stock [32] reconstructed the relative positions of India and Eurasia using revised histories of movements between India and Somalia, and between North America and Eurasia, in addition to the opening of the East African Rift. van Hinsbergen et al. [2] reassessed the relative rates of India–Eurasia motion by combining relative plate motions estimated using marine geophysical data from the Atlantic and Indian Oceans. All of these studies, despite their contrasting methods and variable uncertainty, revealed a broadly similar history of convergence between India and Eurasia since ~60 Ma. The convergence rate of India with Eurasia decreased twice during two short periods: (1) by 40–45% between ~50 and 45 Ma, and (2) by ~30% between ~25 and 20 Ma. Apart from these two short periods, the convergence rate has remained relatively stable (Figure 1, lower left inset). The India–Eurasia collision process should therefore be divided into three stages: an early stage between ~60 and 50 Ma, a middle stage from ~50 to 25 Ma, and a late stage after ~25 Ma.

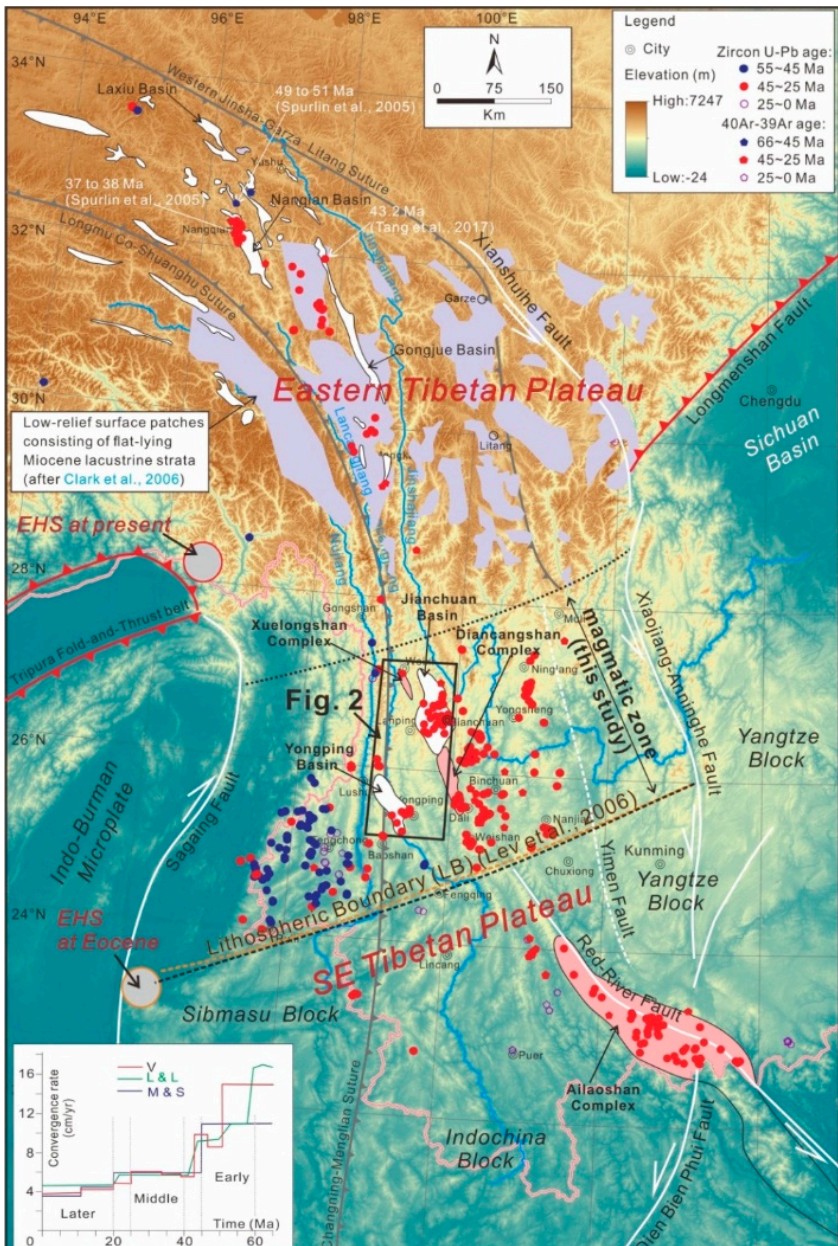

**Figure 1.** Tectonic sketch map of the eastern and southeastern Tibetan Plateau, showing the Cenozoic geology. In the Chinese areas, most Cenozoic plutons and lava have been dated by modern techniques; thus, the distribution of the samples (see Supplementary Table S1 for their GPS locations and data sources) well constrains the distribution of Cenozoic magmatic rocks, which exhibits a NEE–SWW-trending, ~250 km-wide, ~450 km-long zone of intensive magmatic activity (the magmatic zone) where the Yongping basin is located. The northern boundary (dark dotted line) of this zone separates the regions of distinctive Miocene sedimentary geology, whereas its southern boundary (dark dashed line) closely matches the LB (yellow dashed line) of [17] and divides the regions of different Eocene geology (white swaths denote the Eocene contractional basins). White curves are major active lateral sliding faults, whereas red ones denote major active thrusts. Inset on the lower left is the diagram showing the northward shift history of the India plate. Data sources: V, van Hinsbergen et al. [2]; L & L, Lee and Lawver [31]; and M & S, Molnar and Stock [32]. EHS: Eastern Himalaya Syntax.

The second slowdown in convergence rate is relatively less obvious (inset of Figure 1). However, an earliest Miocene transition in the pattern of crustal deformation of the orthogonal belt reinforces the above division. Geological and borehole data indicate that two vast

lake-basins of 100,000 km$^2$ and >50,000 km$^2$, respectively, developed in central Tibet since 23.5 Ma [33]. Meanwhile, a >1300 km-long, <20 km-wide deep meromictic lake, the Kailas Lake, formed along the India–Eurasia suture zone regions during 26–18 Ma probably in an extensional tectonic setting ([34] and references therein). On the contrary, the Eocene basins of central and eastern Tibet are generally found in synclinal troughs in the footwalls of reverse faults of limited throw [35]. They formed when the crust underwent progressive shortening [32,35–39]. The Eocene sedimentary sequences of these syncontractional basins are unconformably overlain by the flat-lying Miocene lacustrine strata [32,36,37,39]. These results indicate that the crustal shortening of the Tibetan Plateau, especially in its central and northern portions, ceased prior to ~23 Ma, and then the crustal shortening migrated to the northern [40] and southern flanks of the plateau. Along the southern flank, the most classic example of coupled tectonic system of "faults/shear zones acting contemporaneously but with opposite kinematics in a collisional belt" started to develop at ~23 Ma [41]. This coupled tectonic system consists of the normal-sense, top-to-the-N or -NE South Tibetan Detachment and the contractional top-to-the-S or -SW Main Central Thrust [42]. In addition, Guo and Wilson [43] revealed a transformation in the nature of the post-collisional magmatism in Tibet at ~25 Ma by comprehensive compilation and modeling simulations of available geochemical data of the Cenozoic magmatic rocks across the Tibetan Plateau. They found that the continental lithosphere-derived components in the mantle sources of the post-collision magmatic rocks are distinctive before and after 25 Ma.

The observations outlined above suggest that two tectonic transitions in the early Eocene and in the earliest Miocene, respectively, divide the tectonic evolution of the Tibetan Plateau into three stages. The nature of the transitions is the key to understanding the dynamics of the India–Eurasia collision [2,28–30,32,44,45]. The long-lived "steady-state" nearly S–N compressional regime of the orthogonally collisional Tibet-Himalaya has left little in the way of diagnostic information regarding these transitions, but such information is probably recorded by the spatial and temporal variation in crustal deformation of the obliquely collisional SE Tibetan Plateau, given its northwards growth [4,46].

Recently, we identified an Eocene basin in SE Tibetan Plateau. We here name it the Yongping basin. The metamorphosed and deformed rocks of this basin were previously incorrectly interpreted as the Late Precambrian or Early Cambrian crystalline basement of the Indochina block [47]. The formation and deformation of the Yongping basin, combined with available geophysical and geological data, may shed light on the deformation history of the oblique collisional belt and then provide positive constraints on the natures of the tectonic transitions. In this paper we present our new observations and structural analyses of the Yongping basin. The timing of its deformation was determined by zircon U-Pb dating of syntectonic leucogranites and step-heating $^{40}$Ar/$^{39}$Ar analyses of muscovite. We use these data, combined with published data, to discuss the dynamics and processes of the India–Eurasia collision.

## 2. Cenozoic Geology of the Eastern and Southeastern Tibetan Plateau

### 2.1. Tectonic Framework

The Eastern and Southeastern (SE) Tibetan Plateau is bounded by the Tripura Fold-and-Thrust Belt [48] to the west and the Longmenshan Fault, the Xiaojiang–Anninghe Fault [49,50], and the Red–River Fault to the east [5] (Figure 1). West of the Tripura Fold-and-Thrust Belt is the Indo-Gangetic Plain, a foreland basin in front of the Main Frontal Thrust of the Himalaya [51]. The Indo-Burman microplate is the hanging wall of the eastward-dipping Tripura Fold-and-Thrust Belt, along which the India lithosphere is being underthrust obliquely towards the east beneath the Indo-Burman microplate [48,52,53]. The eastern boundary of the Indo-Burman microplate is the Sagaing Fault, which is a major right-lateral strike-slip continental fault that extends for >1200 km and connects to the Andaman spreading center at its southern termination [54].

The blocks or terranes east of the Sagaing Fault compose the southwestern margin of Eurasia, and from west to east they are the Tengchong, Baoshan–Sibumasu, Indochina, and Yangtze blocks. These blocks sutured due to the closure of the Paleotethys (e.g., [55]). Voluminous Cretaceous granitoid plutons along the western margins of the Tengchong and Baoshan–Sibumasu blocks form a continuous magmatic belt that is considered the southern equivalent of the Gangdese Belt in southern Tibet [56]. The boundary between the Yangtze and the Indochina blocks is the Ailaoshan–Red River ductile shear zone [25,57], whereas the Indochina is separated from the Baoshan-Sibumasu by the Chongshan–Biluoxueshan ductile shear zone [58]. The northern half of the Indochina block is commonly termed the Lanping–Simao block, and it comprises Cretaceous to Paleocene red beds and underlying Permian-Triassic volcaniclastic rocks [55]. The geology of the Lanping–Simao block, where the Yongping basin is located, is shown in Figure 2.

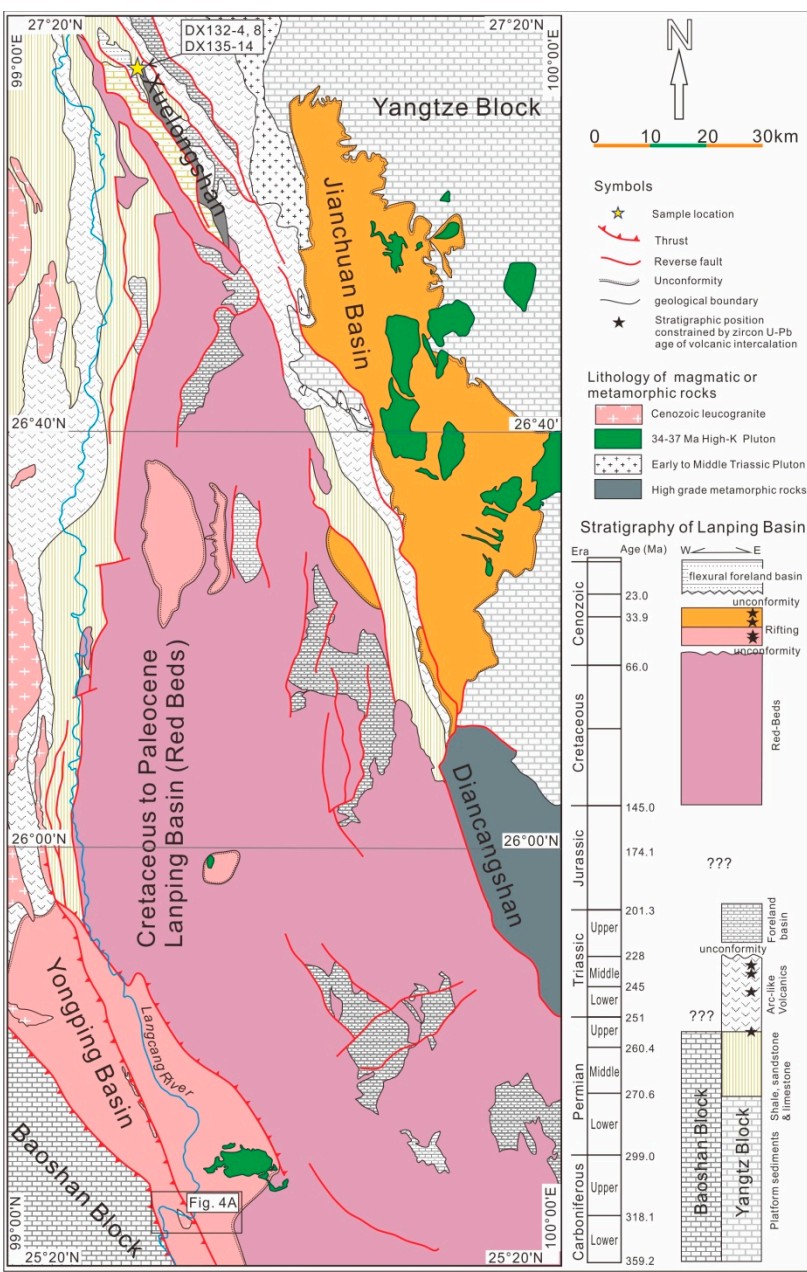

**Figure 2.** Geological map of the Cretaceous to Paleocene Lanping basin and its adjacent regions. The Lanping basin was partly unconformably covered by the Eocene Yongping basin. The location of Figure 2 is shown on Figure 1.

*2.2. A New Cenozoic Tectonic Division*

Numerous Cenozoic basins, in addition to a large amount of small-scale Cenozoic plutons, were identified in the 1970s and 1980s by geological mapping of the eastern and southeastern Tibetan Plateau on the scale of 1:200,000 [47]. Accumulation of geochronological data of modern technologies reveals the spatial and temporal distribution of these Cenozoic units that helps us to suggest a new tectonic division of the eastern and SE Tibetan Plateau (Figure 1).

A comprehensive compilation of recently published geochronological data (see Supplementary Table S1 for dating results and GPS locations of the dated samples) demonstrates that >50% Cenozoic plutons concentrate in a NEE–SWW trending, >450 km-long, ~250 km-wide magmatic zone (Figure 1). The magmatic zone terminates in the Sagaing Fault and in the Yimen Fault to the west and east, respectively. The southern boundary of this zone is spatially coincident with the geophysical LB [13,17]. The northern boundary roughly represents the southwestward extending line of the Longmenshan Fault, although they are slightly offset by the left lateral Xiaojiang-Anninghe Fault. Published zircon U-Pb data demonstrate that the ages of the plutons within the NEE-trending zone (Figure 1) are gradually younger from 55–35 Ma in the west [59–61] to 37–34 Ma in the east [62–65]. In addition to the magmatic zone, a few Late Eocene plutons concentrate in a small triangular region along the southern segment of the brittle Red River Fault [64,66].

The Cenozoic basins were previously assigned a Miocene age based entirely on biostratigrapic and lithostratigraphic correlation [47]. New geochronological data [35–37,67–73] demonstrate that many of them, such as the Nangqian, Laxiu, Mankang, Gongjue, and Jianchuan basins (Figure 1), in the regions eastern of the EHS, developed during the Eocene. They are mostly filled by coarse-grained reddish sandstones with numerous pebble-horizons and minor volcanic intercalations that formed in a contractional setting during 51–34 Ma. In addition, these Eocene basins developed solely in the regions north of the LB. No synchronic basin was, however, positively identified in the regions south of the LB (Figure 1).

Since 23.5 Ma, two vast lake-basins developed in central Tibet. They were dominated by light-grey fine-grained siliciclastic rocks of the Wudaoliang Formation [33]. The Miocene lakes also extended to the eastern Tibet, which now shows a mosaic of extensive, low-relief, high-elevation landscape patches in the outcrops (Figure 1) [67,74,75]. These patches represent a preexisting, low-relief landscape [76]. In central and eastern Tibet, the flat-lying Miocene lacustrine strata unconformably underlie the Eocene sedimentary sequences of syncontractional basins, such as the Nangqian and Laxiu basins [32,36,37,39]. It is interesting that the northern boundary of the magmatic zone represents the southernmost termination of the region where the relatively undeformed, flat-lying Miocene lacustrine strata are exposed (Figure 1). The vast Miocene lake-basins did not extend into the SE Tibetan Plateau where, alternatively, numerous isolated, small-scale, transpressional basins are preserved in synclinal troughs in the footwalls of reverse faults of limited throw. Their fillings are mainly coarse-grained reddish conglomerates (e.g., [73,77]).

Based on the tempo-spatial distribution of the Cenozoic geologic units outlined above, the eastern and SE Tibetan Plateau, i.e., the regions east of the EHS, can be subdivided into three segments (Figure 1): (1) The north segment refers to the regions north of the northern boundary of the magmatic zone, which is the eastern Tibetan Plateau. This segment exhibits the same Eocene and Miocene geology as that of central Tibet. (2) The middle segment is the NEE–SWW-trending, ~250 km-wide magmatic zone, which records the same Eocene but different Miocene sedimentary geology as central and eastern Tibet. (3) The south segment refers to the regions south of the NEE-trending magmatic zone. This segment is characterized by distinctively different Eocene and Miocene geology in contrast to that of central and eastern Tibet, but the Miocene geology seems to be the same as that of the magmatic zone. Thus, the southern and northern boundaries of the magmatic zone separate the regions with different tectonic history during the Eocene and the Miocene, respectively, probably resulting from the northward growth of the oblique collision belt.

## 3. Geology of the Yongping Basin

The newly identified Yongping Basin is located in the central part of the NEE-trending magmatic zone (Figure 1). Our field studies and sampling were conducted along the two highways that line the northern and southern banks of the Lancang (Mekong) River (Figure 2). These highways were reconstructed during our field work, and the outcrops were therefore mostly fresh and reasonably continuous. The sedimentary rocks along these highways have experienced heterogeneous deformation and metamorphism so that their depositional age was incorrectly interpreted to be Precambrian or Early Paleozoic [47]. Some deformed granite intrusions and their metamorphic wall-rocks were interpreted to represent the intracontinental Chongshan-Biluoxueshan ductile shear zone [58]. Our new sedimentary, structural, geochronological, and thermochronological data outlined below lead to new interpretations.

### 3.1. Sedimentary Geology

We estimate the total thickness of the exposed Eocene sedimentary succession of the Yongping basin to be >2500 m (Figure 3A). The fills of this basin unconformably overlie the Cretaceous to Paleocene red beds, characterized by a ~2 m-thick light-gray clast- or matrix-supported basal conglomerate at the bottom. The conglomerate consists of subrounded pebbles and granules of sandstone, mudstone, and limestone in a poorly sorted matrix. The conglomerate grades upwards into fine-grained turbidites that show three cycles with different rocks assemblage and sedimentary textures.

The lower cycle consists of light-gray, massive- to thick-bedded, medium- to coarse-grained litho-feldspatho-quartzo sandstones [78] in the lower part (~250 m thick) and siltstones with two sandy limestone intercalations (each ~2 m thick) in the upper part (~150 m thick). The sandy limestones consist of carbonate extraclasts. Irregular erosive bases, normal grading, and parallel laminations are common in the coarse-grained sandstones, and ripple laminations are well preserved locally in the siltstones. We interpret these turbidites as a Bouma sequence [79] of Bab and Babc, representing the underwater upper-fan succession overlying the debris flow deposits. Tuff-bearing horizons are common in the outcrops, which contain abundant biotite crystal fragments (Figure 3B).

The middle cycle (Figure 3A) consists of ~120 m-thick light-gray massive- to thick-bedded, medium-grained litho-feldspatho-quartzo sandstones at the lower portion, ~40 m of gray fine-grained litho-feldspatho-quartzo sandstones in the middle, and ~30 m of dark-gray muddy siltstones and mudstones in the upper portion. A 2–10 m-thick limestone occurs at the top. These rocks contain thin biotite-rich ash-fall tuff layers and abundant biotite crystal fragments. Normal grading, erosive bases, wavy ripples, and parallel laminations (Figure 3C) are preserved in some outcrops, indicating a turbidite depositional environment. They consist mainly of the Bouma sequence of Bab, Babc, and Bcde. In contrast to the lower cycle, the litho-feldspatho-quartzo sandstones in the middle cycle are less voluminous and a higher proportion of fine-grained turbidite dominates this cycle. In addition, the sediments are relatively well sorted and have higher compositional maturity. These results demonstrate that the middle cycle probably deposited in the channel environment of the middle fan setting.

The upper cycle with a minimum thickness of 1700 m is mainly composed of fine litho-feldspatho-quartzo sandstones, siltstones, and shales (Figure 3D,E), with minor thin limestone and chert beds (Figure 3F,G). The fining-upwards Bouma sequence of Bcde is common, with minor channel deposits of Ba. Erosional basal surfaces (Figure 3F,G), normal grading, and load structures are locally preserved. We interpret this cycle to be deposited in the deep-water basin setting.

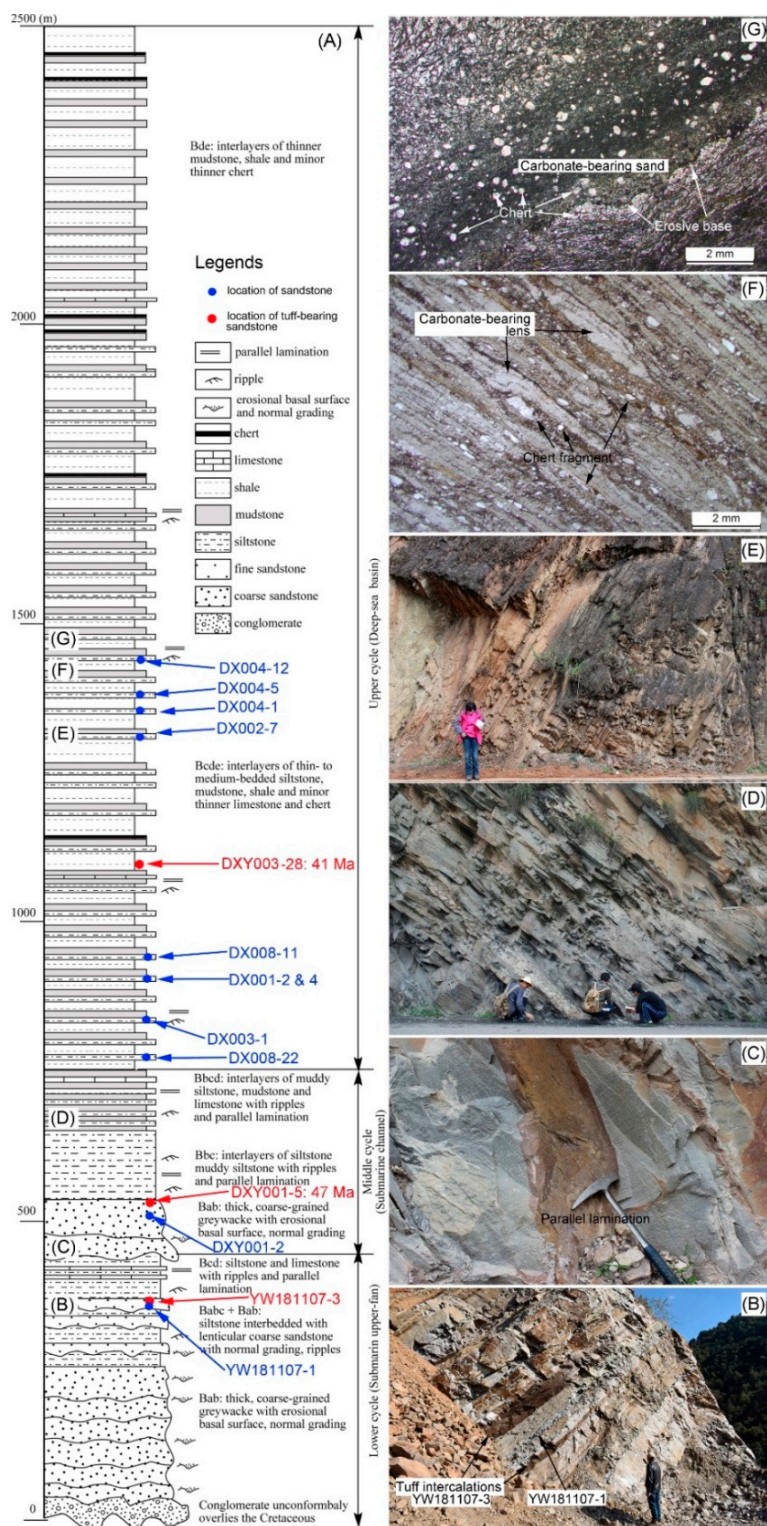

**Figure 3.** (**A**) Schematic Eocene stratigraphic column of the Yongping basin. (**B**) Field photograph of a tuff horizon of predominantly biotite crystals intercalated with coarse to medium-grained sandstone horizons. (**C**) Parallel laminations in massive sandstone. (**D**) Thinly bedded fine sandstone intruded by an undeformed leucogranite dike. (**E**) Flysch sequence consisting of fine sandstone and mudstone. (**F,G**) Photomicrographs (plane polarized light) of fine-grained turbidite show chert fragments and carbonate-bearing lens-shaped bodies. Erosive surfaces are common. The blue and red dots in (**A**) show the stratigraphic locations of detrital zircon and volcanic samples, respectively; the stratigraphic locations of (**B–G**) are also shown in (**A**).

### 3.2. Magmatism

Three relatively larger leucogranite intrusions were identified along a ~10 km-long section (Figure 4A). The biggest one has an exposed width of 1.4 km in the E–W direction. The biggest intrusion was emplaced in the upper deep-sea basin succession, whereas the other plutons intrude the middle- or lower-cycle successions. The host rocks of the biggest leucogranite intrusion metamorphosed heterogeneously, displaying zoned contact metamorphic aureoles. A small syenite intrusion with a diameter of <20 m was emplaced in the northeastern part of the section.

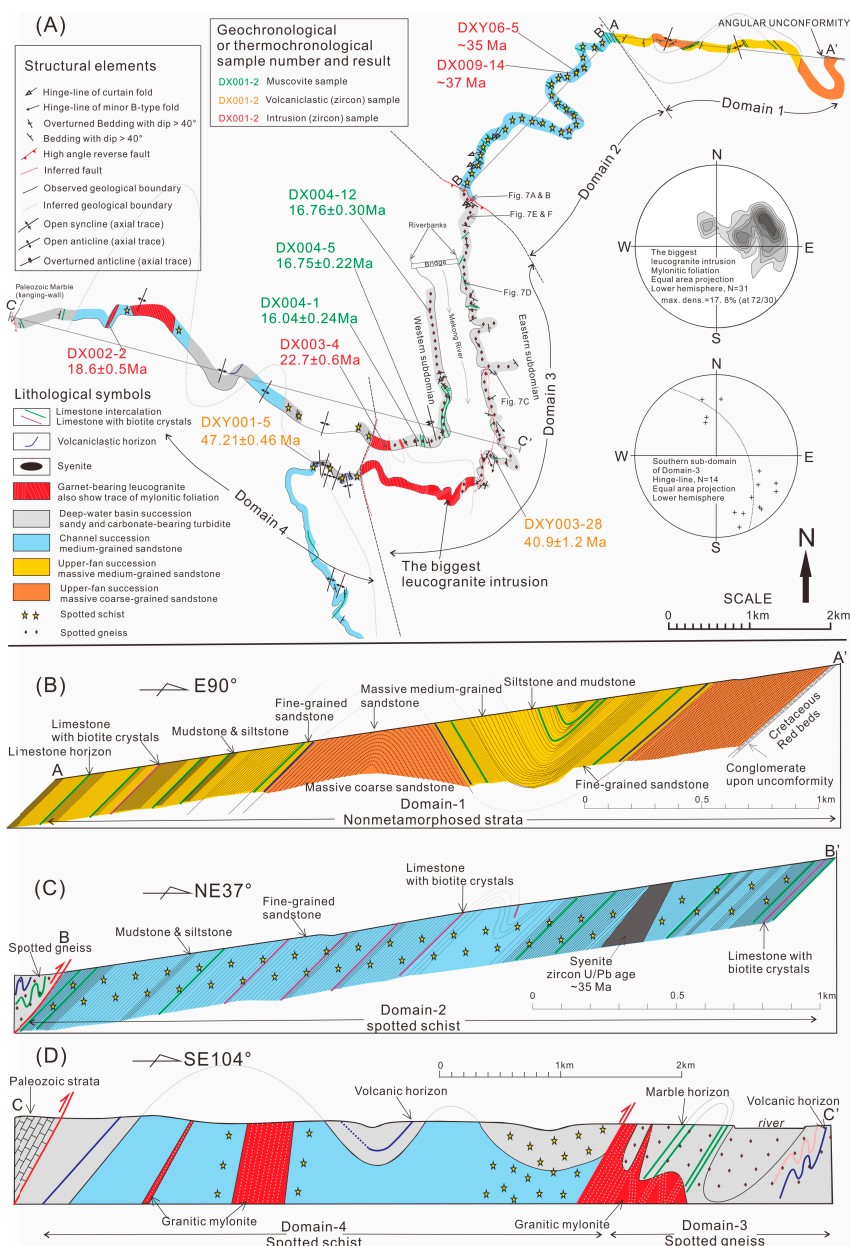

**Figure 4.** (**A**) Geological map of the highways along the northern and southern banks of the Mekong River that shows the locations and dating results of samples collected for geochronological and thermochronological analyses. See Figure 2 for its location. Insets on the central right and lower right are stereographic projections (lower hemisphere) of poles to the mylonitic foliations in the biggest leucogranite and of the hinge lines of small-scale B-type folds in the eastern sub-domain of Domain 3, respectively. (**B–D**) are structural profiles without vertical exaggeration across Domains 1, 2, and 3 and 4, respectively. Their locations are shown in (**A**).

The leucogranites are light gray and consist predominantly of feldspar and quartz and minor oriented tourmaline needles. Euhedral garnets are widespread although their modal content is not higher than 3% (Figure 5A). Of note, the relatively larger leucogranites that occur in outcrops greater than 10 m in width have all been ductilely deformed, whereas leucogranite dikes that are less than 1 m wide are undeformed (Figure 5A). The syenite consists of K-feldspar, plagioclase, clinopyroxene, and biotite, which also have not been deformed (Figure 5B).

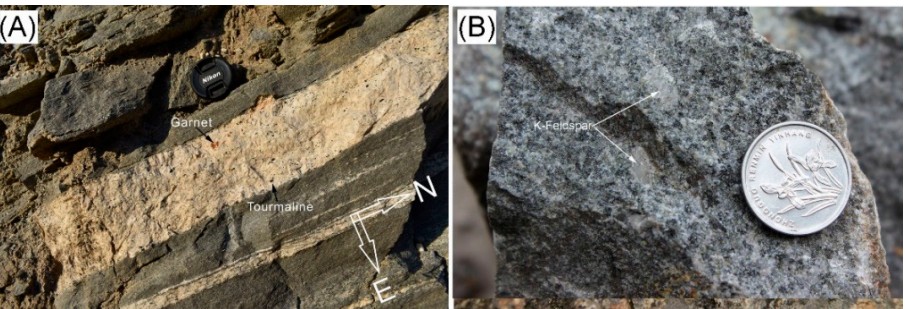

**Figure 5.** Field photographs of a leucogranite dyke (**A**) and syenite (**B**). Neither shows any evidence of deformation.

### 3.3. Structural and Metamorphic Geology

3.3.1. Structures and Metamorphism of the Sedimentary Rocks

The sedimentary rocks were heterogeneously deformed and metamorphosed, and show varieties of structural patterns and metamorphic mineral assemblage. Moreover, the different structural patterns seem to be coupled with distinctive mineral assemblages (see following sections for details). We divided the section along the highways into four domains according to structural associations and interference patterns (Figure 4A), which are summarized in Figure 6 and are documented sequentially from the northeast to southwest.

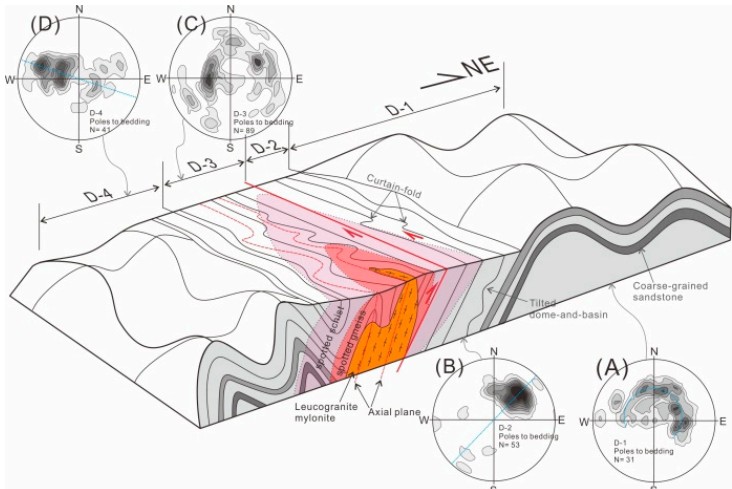

**Figure 6.** Conceptual diagram that summarizes the structural interference patterns of each domain (D-1 to D-4) and their spatial distribution of the Yongping basin. (**A–D**) are the stereographic projections of poles to bedding (lower hemisphere) for Domains 1 to 4, respectively.

Domain 1 contains the lower cycle rocks that were not significantly affected by metamorphism. The rocks are characterized by well-preserved primary minerals and sedimentary textures (Figure 3B,C). Repetition of various sedimentary beds, particularly the sandy limestone intercalations, together with changes in the bedding dips (Figure 4A), defines an open anticline and an open syncline in profile view (Figure 4B). Stereographic projections of poles to beddings form a small circle on a Schmidt steronet (Figure 6A), indicating

the folds are conical in shape. In map view the folds display a dome-and-basin pattern (D-1, Figure 6).

Domain 2 is located to the west of the dome-and-basin domain and contains the middle cycle rocks. Most pelitic rocks in Domain 2 have been metamorphosed to spotted mica schist with large biotite poikiloblasts set in a matrix of very fine biotite, muscovite, quartz, and accessary minerals, displaying a diagnostic texture of contact metamorphism. The poikiloblasts are crowded with tiny inclusions of the matrix phases (Figure 7A,B). The preferred orientation of fine biotite and muscovite defines a foliation in the matrix of the schist, whereas the biotite porphyroblasts are randomly oriented. The foliations are broadly parallel to the beddings. On a bedding surface, both poikiloblasts and their matrix phase minerals are randomly oriented.

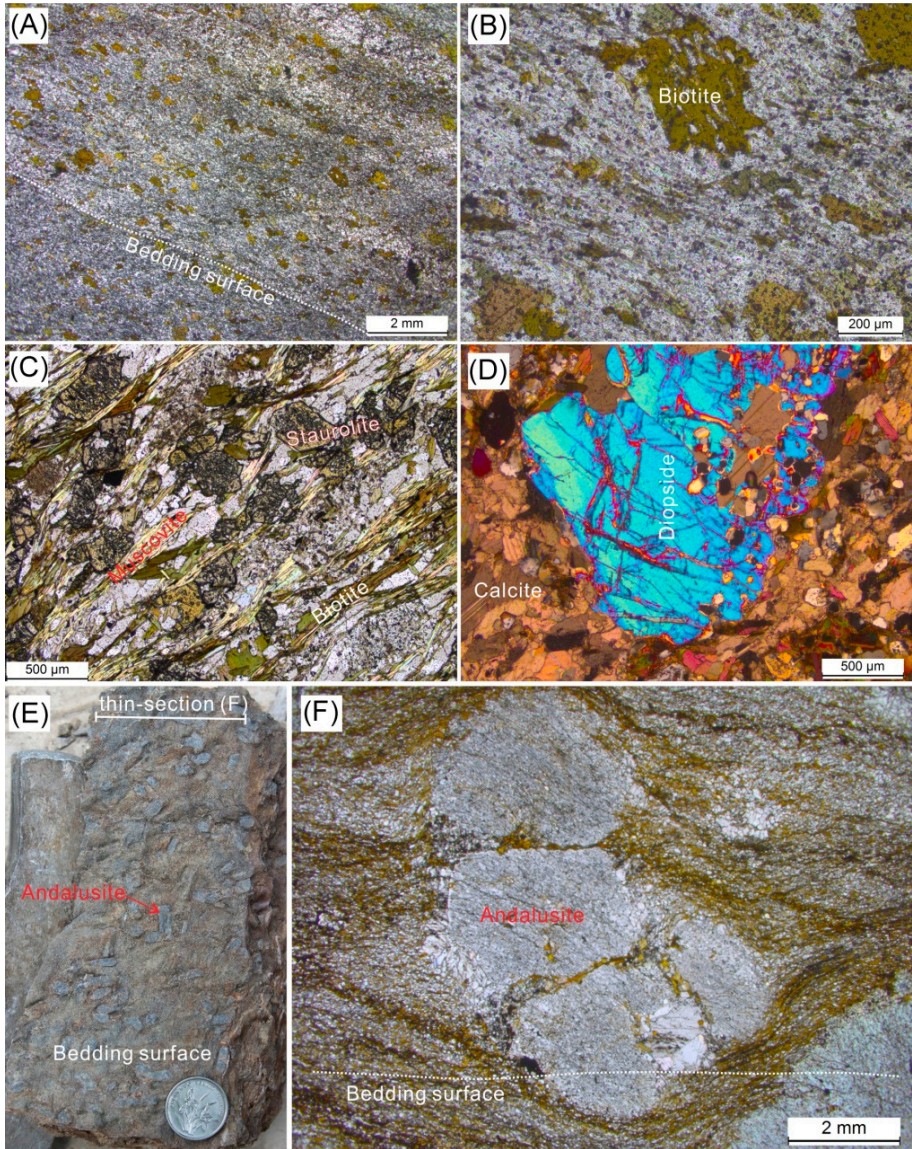

**Figure 7.** Photomicrographs of contact metamorphosed and deformed rocks. (**A**,**B**) Spotted schists in which the porphyroblasts are entirely biotite (plane polarized light). (**C**–**F**) Spotted gneiss in which the porphyroblasts vary from staurolite ((**C**), plane polarized light) to diopside ((**D**), cross polarized light) or andalusite ((**E**,**F**), plane polarized light) depending on the bulk composition of the rock. In a bedding surface, the elongate porphyroblasts are randomly oriented (**E**). See Figure 4A for the location of each sample.

Field observations and structural measurements reveal that most beds dip to the southwest at angles of 40° to 50° (Figures 4C and 6), as shown by stereographic projections of poles to beddings (Figure 6B). A few meter-scale intra-bedding curtain folds [80] display highly variable geometries, which are limited to the thinly bedded fine-grained sandstones. These folds are mostly asymmetrical, isoclinal to recumbent, and disharmonic, and they commonly decrease in amplitude and fade out laterally. These folds verge consistently towards the northwest, indicating top-to-the-northwest shearing (Figure 8A). In contrast, some folds are open and symmetrical, exhibiting a tilted dome-and-basin pattern (Figure 8B,C), and a few display patterns intermediate between the highly asymmetrical and symmetrical end-member types. The beds adjacent to these folded horizons (both above and below) do not show folds (Figure 4C). The hinge lines and axial planes of the asymmetrical curtain folds are subparallel to the dips of bedding in the adjacent layers (Figure 4A).

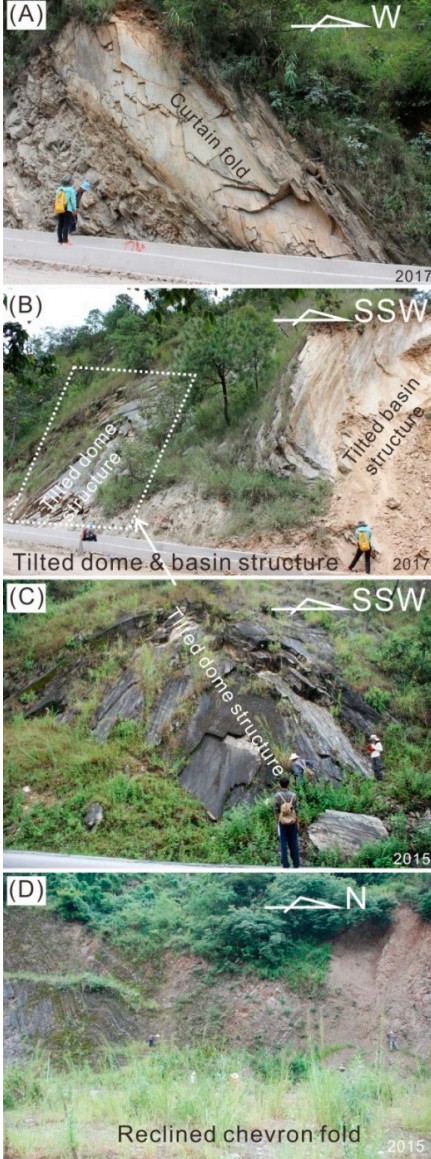

**Figure 8.** Field photographs showing (**A**) a reclined curtain fold, (**B**) a tilted dome-and-basin structure, (**C**) the dome of the dome-and-basin structure of (**B**) in Domain 2, and (**D**) the reclined chevron folds that superposed upon the earlier stage overturned folds. Note that the pictures of (**B**,**C**) were taken at different times, so that they do not exactly match.

The western boundary of Domain 2 is a southwestwards-dipping reverse fault (Figure 4A), along which the folded and metamorphosed deep-water basin succession has been thrust over the channel succession. The regions southwest of the reverse fault and northeast of the biggest leucogranite intrusion define Domain 3 (Figure 4A; D-3 in Figure 6). The pelitic beds of this domain have been metamorphosed to spotted gneiss. The gneiss consists of biotite, muscovite, staurolite (Figure 7C), diopside (Figure 7D), andalusite (Figure 7E,F), sillimanite, and locally, cordierite. The andalusite and staurolite both occur as spotted poikiloblasts that are crowded with tiny inclusions of the matrix phases.

In spite of the relatively high-grade metamorphism, sedimentary beddings are still well preserved, which is a common feature of contact metamorphosed rocks. The trends and dips of bedding in this domain are highly variable, and the structures are complex. The spatially variable structures of this domain seem to be closely related to the shape of the biggest leucogranite (Figure 4A). Two subdomains are subdivided accordingly. The western bank of the Mekong River that lies in the north of the biggest leucogranite defines the western subdomain, which displays an overturned syncline to the west and an overturned anticline to the east (Figure 4D). The latter is partly destroyed by the Mekong River. The folds are both one hundred meters wide, north–south trending, and eastwards-verging. E–W trending, open to tight small reclined chevron folds have superimposed upon them (D-3, Figures 6 and 8D). The eastern subdomain is located in the east of the largest leucogranite, and it features numerous small-scale overturned to recumbent folds and high-angle reverse faults (Figure 4A). These small-scale folds are mostly eastwards-verging. Projections of their hinge lines form a great circle on a stereograph (Figure 4A, inset), suggesting the hinge lines plunge within a SWW-dipping plane although they appear to be randomly oriented in map view.

The regions west of the biggest leucogranite intrusion belong to Domain 4. Most pelitic rocks of the middle and upper successions in Domain 4 have been metamorphosed to spotted mica-schist, which are similar to those of Domain 2. Repetition of strata and changes in the dips of bedding define open synclines alternating with anticlines (Figures 4D and 6D). The axial planes of these folds are nearly vertical with wavy trends in map view. Their hinge lines are horizontal and wavy in map view, exhibiting an elongate dome-and-basin pattern (D-4, Figure 6). The long axials of the elongate basin-and-dome strike NNE–SSW (Figure 6D).

In summary, spotted gneisses were developed exclusively to the north and east of the biggest leucogranite body. The gneiss and the biggest intrusion are surrounded by spotted schists (Figure 4), showing an aureole of contact metamorphism. The development of both andalusite and sillimanite suggests that the spotted gneiss formed under the *P–T* conditions of >500 °C and <3 kbar [81]. The spatial distribution of the metamorphic zones in the aureole is unusual, and is probably related to the syn-intrusion deformation.

3.3.2. Structures of Leucogranites

The leucogranite intrusive bodies, with the exception of the thin dykes, show evidence of mylonitization (Figure 9A–C). The biggest leucogranite intrusion exhibits a complicated structural pattern, whereas the structures of the other smaller intrusive bodies are relatively simple. The latter contain S-type mylonites (i.e., mylonites that are characterized by a simple foliation and no stretching lineation [82]) with foliations of constant strike and dip, although the strikes and dips vary slightly among the intrusions (Figure 4A). The biggest intrusion displays an N–S striking, wavy but smooth western boundary, and a highly irregular eastern boundary. The mylonitic foliation in a ~700 m-wide zone along the western half of the biggest intrusion has a constant dip towards the west. Shape-fabric analyses on 20 oriented hand specimens reveal that the mylonites are S-types, so that all foliation-perpendicular thin sections, including dip-parallel or dip-perpendicular surfaces, display the same shape fabrics, as exemplified by the same aspect ratios of the feldspar porphyroclasts (Figure 10). Asymmetrical textures of these sections indicate an equal development of dextral and sinistral shearing. By comparison, three ductile shear zones

in the eastern half of the biggest leucogranite intrusion interconnect with each other to form a network showing strain partitioning into anastomosing high-strain zones (mylonite) and elongated low-strain domains (protomylonite). These shear zones contain foliations with different strikes and dips (central-right inset of Figure 4A), and very rare horizontal stretching lineations are locally observed.

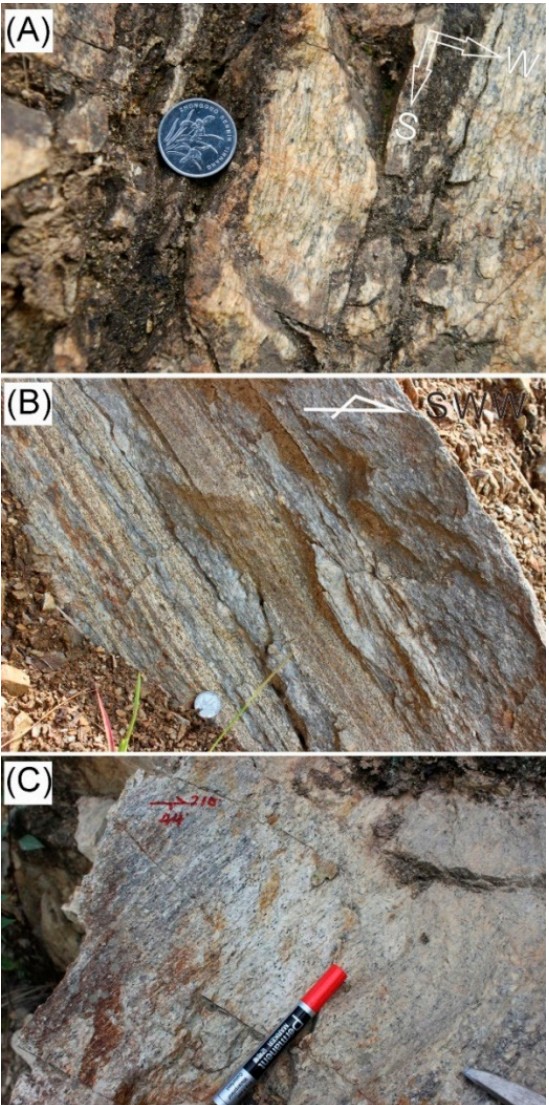

**Figure 9.** Field photographs of mylonitized leucogranite. (**A**,**B**) are the views of the horizontal and the vertical surfaces, respectively. (**C**) is the view of a mylonite-foliation-parallel surface. No stretching lineation is evident but slickensides developed on the foliation.

The strain of the leucogranite is mainly concentrated in the quartz whereas the feldspar grains are weakly deformed (Figure 10A–D). The boundary between the highly stretched quartz and weakly deformed feldspar grains is generally sharp and smooth. Quartz grains form ribbons that wrap around the slightly elongate feldspar grains. New grains within the quartz ribbons show highly irregular boundaries (Figure 10B,D), suggesting that grain boundary migration (GBM) took place as the dynamic recrystallization mechanism under high-temperature conditions (500–700 °C; [80]). The estimated temperature of deformation is comparable to that of the contact metamorphism, as indicated by the coexistence of andalusite and sillimanite (>500 °C) [81].

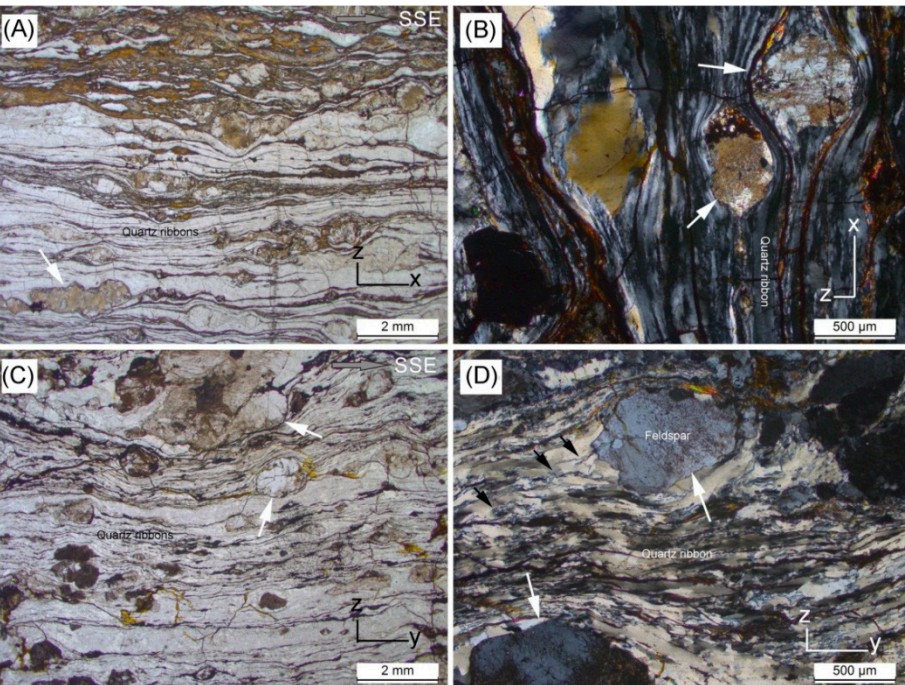

**Figure 10.** (**A,C**) Photomicrographs (plane polarized light) showing the mineral shape fabrics of the mylonitized leucogranite in foliation-dip-perpendicular and foliation-dip-parallel thin sections, respectively. (**B,D**) Photomicrographs (cross polarized light) of parts of (**A,C**), respectively. Note their different scales. All the porphyroclasts are feldspar with a sharp and smooth boundary (indicated by white arrows), and they are surrounded by quartz ribbons. The quartz ribbon consists of very fine new grains with highly irregular boundaries (indicated by dark arrows).

## 4. Geochronology and Thermochronology

### 4.1. Sampling and Methodology

In order to determine the timing of the deposition, magmatism, metamorphism, and deformation of the Yongping basin, 14 sedimentary and/or volcaniclastic samples were collected for zircon separation and U-Pb dating using LA–ICP–MS techniques. These samples include one coarse-grained sandstone and one biotite-rich ash-fall tuff from the lower cycle, one medium-grained sandstone and one volcaniclastic sample from the middle cycle, and ten spotted schists or gneisses from the upper cycle (see Figure 3A for their stratigraphic locations). The protolith of one spotted schist sample is volcaniclastic rock according to results of the field and petrographic studies. Three fresh samples were collected from lamprophyre dikes for biotite separation to undertake step-heating $^{40}$Ar/$^{39}$Ar analyses. These dikes intruded the Permian–Triassic rhyolite in the basement of the Yongping basin ([83]; see Figure 2 for their locations), which probably represent the volcanic conduits of the biotite ash-fall tuffs. In addition, two leucogranite and two syenite samples were also collected for zircon separation and LA–ICP–MS U-Pb dating (see Figure 4A for their locations). Three two-mica spotted schists from the contact aureole were collected to separate muscovite for step-heating $^{40}$Ar/$^{39}$Ar dating (Figure 4A). The analyzed results, GPS locations, and lithologies of the samples are listed in Table 1. See Supplementary Materials for analytical methods. The detailed U/Pb analytical results of the detrital zircons are listed in Supplementary Table S2. Supplementary Table S3 lists the U/Pb analytical results of zircons of plutons. The $^{40}$Ar/$^{39}$Ar analytical results of biotites and muscovites are listed in Supplementary Table S4.

**Table 1.** GPS localities and geochronological/thermochronological results of the samples from the Yongping basin, SE Tibet.

| | Sample No. | Lithology | Total Analyses | Concordant Analyses | Weighted Mean Age | GPS Coordination Latitude | Longitude |
|---|---|---|---|---|---|---|---|
| | | | | | | | |
| | *Samples from the Lower Cycle for zircon LA-ICP-MS analyses* | | | | | | |
| 1 | YW181107-1 | sandstone | 120 | 120 | | 25°27′37.7″ N | 99°23′28.1″ E |
| 2 | YW181107-3 | volcanicalstics | 77 | 74 | Without weighted age | | |
| | *Samples from the Middle Cycle for zircon LA-ICP-MS analyses* | | | | | | |
| 1 | DXY001-2 | sandstone | 80 | 77 | | 25°23′48.7″ N | 99°19′40.8″ E |
| 2 | DXY001-5 | volcaniclastics | 80 | 79 | 47.21 ± 0.46 (*n* = 5, MSWD = 1.2) | 25°24′20.4″ N | 99°19′40.5″ E |
| | *Samples from the Upper Cycle for zircon LA-ICP-MS analyses* | | | | | | |
| 1 | DX001-2 | | 38 | 37 | | 25°25′40.1″ N | 99°17′32.7″ E |
| 2 | DX001-4 | schist | 25 | 23 | | 25°25′40.1″ N | 99°17′32.8″ E |
| 3 | DX002-7 | | 32 | 26 | | 25°25′10.7″ N | 99°19′38.8″ E |
| 4 | DX003-1 | | 42 | 23 | | 25°24′55.6″ N | 99°19′24.2″ E |
| 5 | DXY003-28 | volcaniclastics | 28 | 18 | 40.9 ± 1.2 (*n* = 3, MSWD = 3.1) | 25°24′43.5″ N | 99°20′51.4″ E |
| 6 | DX004-1 | | 32 | 23 | | 25°24′42.8″ N | 99°20′19.0″ E |
| 7 | DX004-5 | | 34 | 23 | | 25°24′42.7″ N | 99°20′19.2″ E |
| 8 | DX004-12 | gneiss | 24 | 19 | | 25°25′22.7″ N | 99°20′19.5″ E |
| 9 | DX008-11 | | 22 | 14 | | 25°24′42.4″ N | 99°20′52.1″ E |
| 10 | DX008-22 | | 28 | 18 | | 25°25′34.7″ N | 99°20′39.8″ E |
| | *Plutonic samples for zircon LA-ICP-MS analyses* | | | | | | |
| 1 | DX002-2 | leucogranite | 27 | 16 | 18.56 ± 0.38 (*n* = 12, MSWD = 3) | 25°25′31.9″ N | 99°18′03.3″ E |
| 2 | DX003-4 | | 30 | 13 | 22.67 ± 0.63 (*n* = 9, MSWD = 1.02) | 25°24′45.5″ N | 99°20′27.8″ E |
| 3 | DXY006-5 | syenite | 17 | 12 | 35.3 ± 1.5 (*n* = 4, MSWD = 11.5) | 25°27′18.1″ N | 99°21′22.5″ E |
| 4 | DX009-14 | | 13 | 4 | without WMA | 25°27′18.1″ N | 99°21′22.5″ E |
| | *Samples for* $^{40}Ar/^{39}Ar$ *analyses* | | | | | | |
| 1 | DX004-1 | Gneiss (muscovite) | | | 16.04 ± 0.24 Ma | 25°24′42.8″ N | 99°20′19.0″ E |
| 2 | DX004-5 | | | | 16.75 ± 0.22 Ma, | 25°24′42.7″ N | 99°20′19.2″ E |
| 3 | DX004-12 | | | | 16.76 ± 0.30 Ma | 25°25′22.7″ N | 99°20′19.5″ E |
| 4 | DX132-4 | Lamprophyre (biotite) | | | 48.08 ± 0.52 Ma | 27°21′06.1″ N | 99°07′33.9″ E |
| 5 | DX132-8 | | | | 45.70 ± 0.58 Ma | 27°21′06.1″ N | 99°07′33.9″ E |
| 6 | DX135-14 | | | | 36.61 ± 0.42 Ma | 27°21′06.0″ N | 99°07′34.0″ E |

### 4.2. Zircon U-Pb Results

4.2.1. Sandstone and Volcaniclastic Samples

Samples YW181107-1 and YW181107-3 are coarse-grained sandstone and volcaniclastics, respectively, of the lower cycle (Figure 3A). We selected 120 and 74 detrital zircon grains with clear growth zoning from these samples for U-Pb analyses and yielded 114 and 74 concordant results, respectively, with concordance >90% and U/Th ratio mostly <10. With the exception of the youngest $^{206}Pb/^{238}U$ age of 191 Ma, other concordant ages of these two samples range from 201 to 2595 Ma. They exhibit three major groups (Figure 11A): (1) 47.4% of the zircons have ages ranging from 200 to 500 Ma with the peaks at 214, 227, 255, 332, and 434 Ma; (2) 13.0% of the zircons have ages varying from 500 to 1500 Ma, without obvious peaks; and (3) the remaining 39.6% of the zircons are older than 1.5 Ga, with an obvious peak at 1860 Ma and a weaker peak at 2521 Ma.

Samples DXY001-2 and DXY001-5 were collected from the lower portion of the middle cycle (Figure 3A). Eighty detrital zircon grains with clear growth zoning of each sample were dated resulting in 156 concordant U-Pb ages with concordance >90%. Most grains have low U/Th ratios of <10. Six grains from the volcaniclastic sample DXY001-5 have young $^{206}Pb/^{238}U$ ages ranging from 45.9 to 47.6 Ma, and five of them formed a tight cluster with a weighted mean age of 47.21 ± 0.46 Ma (MSWD = 1.2) (Figure 12A). The other

73 zircon grains of this sample and 80 grains from the sandstone sample DXY001-2 are much older, with $^{206}$Pb/$^{238}$U ages of >140 Ma. These 153 analyses on 153 detrital zircon grains display four major groups (Figure 11B): (1) 14.0% of the zircons have ages ranging from 140 to 180 Ma with the peaks at 147 and 158 Ma; (2) 42.9% of the zircons have ages ranging from 200 to 500 Ma with the peaks at 204, 225, 237, 252, 393, and 419 Ma; (3) 16% of the zircons have ages varying from 500 to 1500 Ma, without obvious peaks; and (4) the remaining 27.1% of the zircon grains are older than 1.5 Ga, with two peaks at 1861 and 2526 Ma.

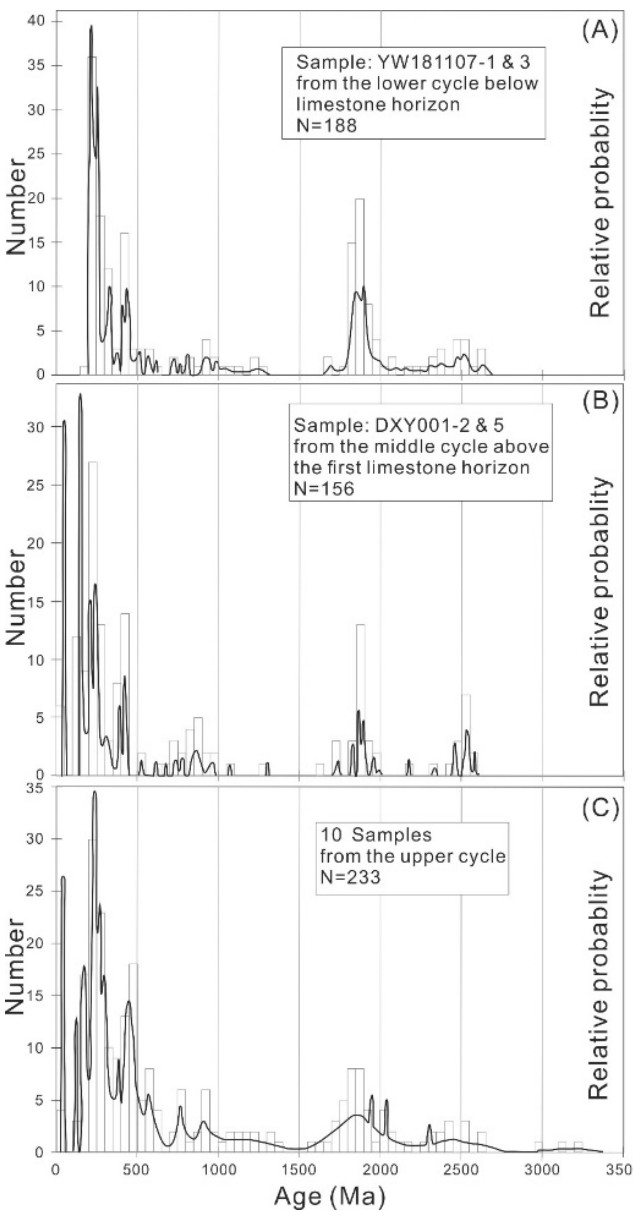

**Figure 11.** Probability density plotting (PDP) with optional histograms of the $^{206}$Pb/$^{238}$U (<1000 Ma) or $^{206}$Pb/$^{207}$Pb (>1000 Ma) ages of zircons in sedimentary rocks of the lower cycle (**A**), the middle cycle (**B**), and the upper cycle (**C**) with bin width = 50 Ma of the Yongping basin, Southeastern Tibetan Plateau. See Figure 3A for the stratigraphic locations of samples.

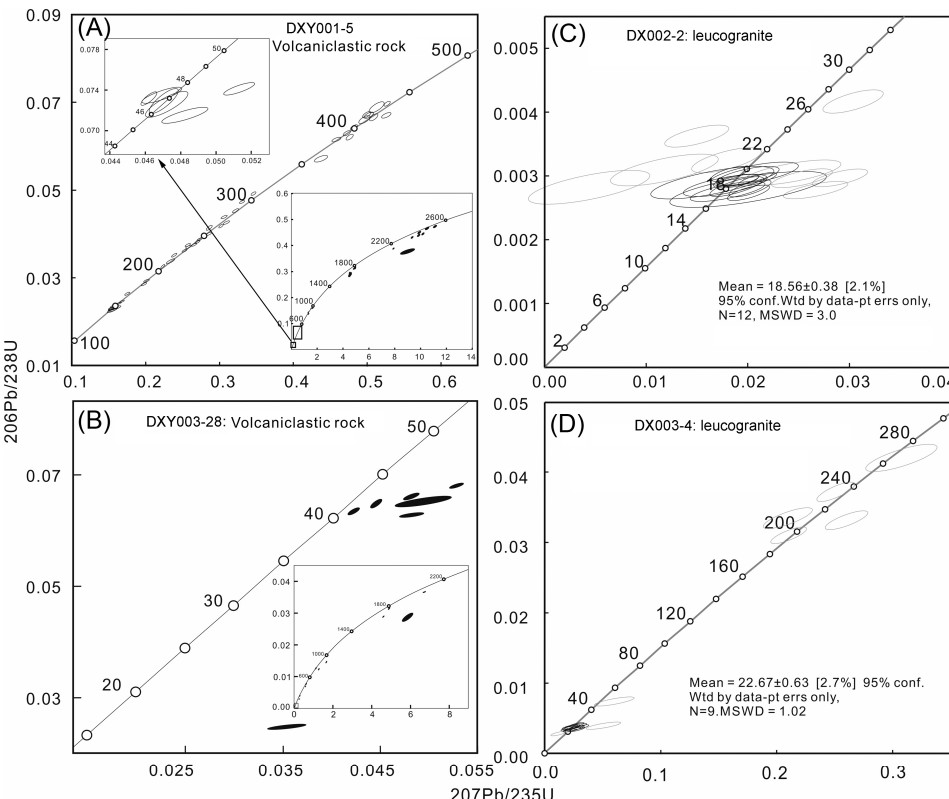

**Figure 12.** Zircon U–Pb concordia diagrams of the volcaniclastic rocks (**A**,**B**) and leucogranites (**C**,**D**) of the Yongping basin, southeastern Tibetan Plateau. See text for a detailed discussion.

Ten metamorphosed siltstone samples were collected from the upper cycles. Their zircon grains are mostly small with a length of <150 μm and width of <50 μm. A total of 305 relatively large zircon grains with clear growth zoning were selected for U-Pb dating, and 224 concordant results were obtained with concordance >90% and mostly low U/Th ratios of <10. Seven zircons from the metamorphic volcaniclastic sample DXY003-28 have apparent ages ranging from 43.7 to 40.4 Ma, and three youngest zircons yield a weighted mean age of 40.9 ± 1.2 Ma (MSWD = 3.1; Figure 12B). The remnant analyses exhibit a similar age histogram (Figure 11C) as that (Figure 11B) of the detrital zircons from the middle cycle, showing four major age groups: (1) 14.8% of the zircons have ages ranging from 110 to 188 Ma with a peak at 168 Ma; (2) 52.3% of the zircons have ages ranging from 200 to 500 Ma with peaks at 238, 270, 292, and 445 Ma; (3) the ages of 10.3% zircon grains range from 500 to 1500 Ma without obvious peaks; and (4) the remaining 22% zircon grains are older than 1500 Ma with two weaker peaks at 1852 and 2443 Ma.

### 4.2.2. Intrusions

The small syenite intrusion was sampled twice in 2015 (DX09-14) and in 2017 (DXY06-5). Thirteen zircon grains were derived from sample DX09-14 and were analyzed, yielding three concordant results with $^{207}$Pb/$^{206}$Pb ages of 2009, 1215, and 3098 Ma. Two grains have disconcordant $^{206}$Pb/$^{238}$U ages of 37 Ma and 38 Ma, respectively. Seventeen zircon grains from sample DXY06-5 were dated and 12 concordant results were obtained. Three concordant analyses yielded $^{206}$Pb/$^{238}$U ages of 36.24, 35.53, and 33.99 Ma. We suggest that the small syenite body exposed in the river section is a branch of the large alkaline plutonic complex exposed ~5 km east of our study area (the Zhuopan pluton; Figure 2), which has a weighted mean zircon U-Pb age of 35 ± 0.3 Ma [84].

Samples DX003-4 and DX002-2 were collected from the biggest leucogranite and a smaller intrusion (Figure 4A), respectively, for zircon separations and U-Pb dating. Most zircon grains from both samples are small with a length of <100 μm and variable aspect ratios between 1 and 3. Sixteen of 27 analyses of 27 zircon grains of sample DX002-2 yielded

concordant results. Six concordant results with the youngest ages form a tight cluster with a weighted mean $^{206}Pb/^{238}U$ age of 18.6 ± 0.5 Ma (MSWD = 0.22) (Figure 12C). Thirty analyses of 30 zircon grains from sample DX003-4 yielded 13 concordant results, six of which formed a tight cluster with a weighted mean $^{206}Pb/^{238}U$ age of 22.67 ± 0.63 Ma (MSWD = 1.02) (Figure 12D).

### 4.3. $^{40}Ar/^{39}Ar$ Step Heating Results

Analyses of biotites derived from three lamprophyre samples show spectra with a slight Ar loss (Figure 13A–C). Their preferred ages (PAs) are 48.08 ± 0.52 Ma (including 45.1% released $^{39}Ar$, MSWD = 1.17), 45.70 ± 0.58 Ma (including 43% released $^{39}Ar$, MSWD = 0.19), and 36.61 ± 0.42 Ma (including 76.9% released 39Ar, MSWD = 0.77). Their isochron ages are the same as the preferred ones, within analytical error.

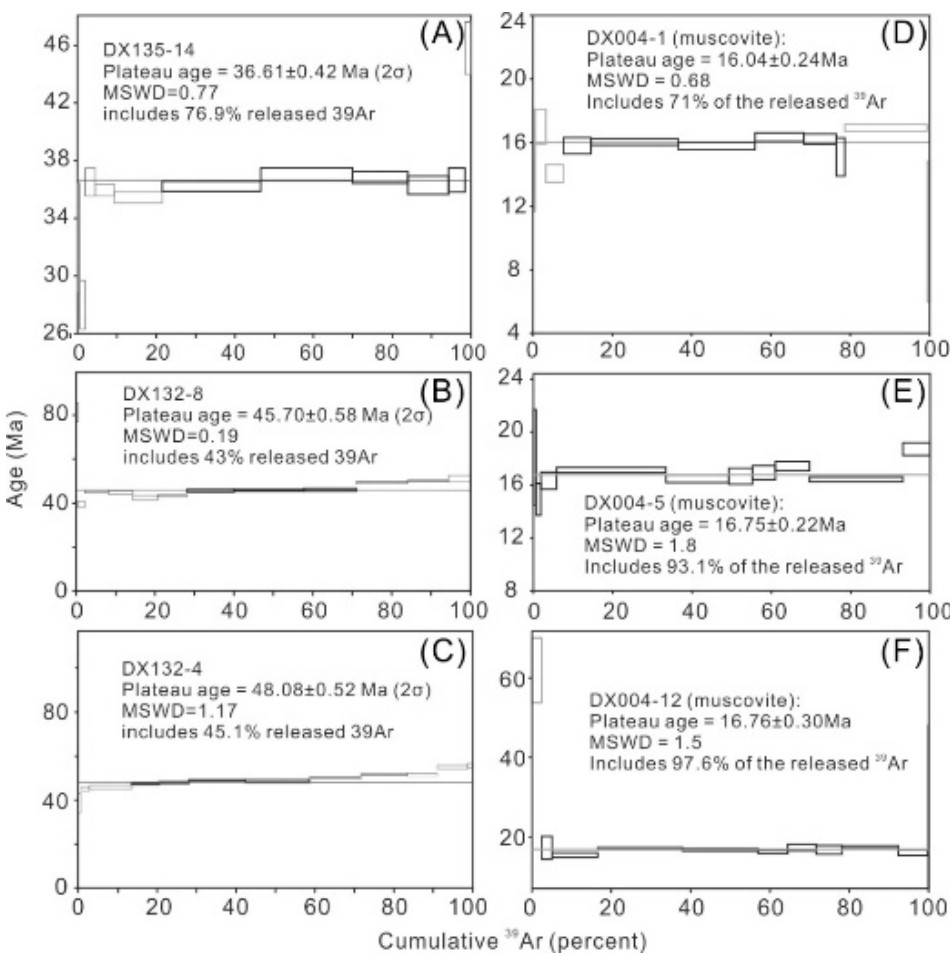

**Figure 13.** $^{40}Ar/^{39}Ar$ age spectra for biotite from the lamprophyre dikes (**A**–**C**) that intruded the basement of the Yongping basin and for muscovite in spotted schist samples (**D**–**F**) from the Yongping basin. The sample locations are marked on Figures 2 and 4A.

Muscovites separated from three two-mica schist samples were subjected to step-heating $^{40}Ar/^{39}Ar$ analyses. Their spectra show no obvious Ar loss (Figure 13D–F) with weighted mean plateau ages (WMPA) of 16.04 ± 0.24 Ma, 16.75 ± 0.22 Ma, and 16.76 ± 0.30 Ma, with MSWD values of 0.66, 1.8, and 1.5, respectively. The WMPAs and isochron ages of each sample are similar, within analytical error.

## 5. Discussion

### 5.1. Timing of Formation and Deformation of the Yongping Basin

Our new data reveal (1) the detrital zircons of the sandstones of the Yongping basin are all older than 120 Ma (Figure 11), but (2) the ages of a few magmatic zircons of the volcaniclastic interlayers are younger than 50 Ma (Figure 12A,B). In addition, (3) the sediments of the Yongping basin were intruded by the ~35 Ma Zhuopan high-Mg alkaline plutons (Figure 2; e.g., [84], this study). These data and cross-cutting relationship demonstrate that the initial deposition of the Yongping basin took place prior to 47 Ma (DXY001-5, Figure 3A) and the deposition lasted to 35 Ma. The crystallized ages of lamprophyric dikes that crosscut the basement of the Yongping basin (Figure 13A–C) are similar to the depositional ages of this basin within analytical error, probably representing one kind of volcanic conduits that likely have fed the multiple interlayers of tuff of the Yongping basin.

The micro-textures (Figure 7) of the spotted schist and spotted gneiss indicate that the contact metamorphism was accompanied by a deformation. Structural studies (Figures 4, 9 and 10) indicate that a strong mylonitization occurred exclusively in the leucogranites under temperatures that were the same (500 to 700 °C) as those during contact metamorphism. Neither the metamorphosed sedimentary rocks (Figure 7) nor the leucogranite dikes (Figure 5A) deformed in a ductile manner, and no strain is recorded by the 35 Ma branch syenite (Figure 5B) or its parental pluton (Figure 2). These observations, taking strain compatibility, which "specifies how strain can vary within a heterogeneously deformed material without causing structural discontinuities, holes between domains, or abrupt changes in the type of strain" [85], into account, suggest that the leucogranites were deformed during or very shortly after their emplacement when their temperature was still higher than 500 °C. The <1 m-thick leucogranite dykes did not deform because of rapid cooling after emplacement. Our zircon U-Pb dating results demonstrate that the leucogranites crystallized during ~23–18 Ma (Figure 12C,D). Our muscovite $^{40}Ar/^{39}Ar$ step-heating results (Figure 13D–F) reveal that the spotted schist cooled to the closure temperature of the K-Ar isotopic system in muscovite (<400 °C; [86]) at around 16 Ma. These data suggest that the rocks of the Eocene Yongping basin were deformed during the period ~23 to 16 Ma. Probably in response to this deformation, widespread deposits of Early Miocene conglomerates were laid down in the SE Tibetan Plateau (e.g., [73,77]).

### 5.2. Formation of the Yongping Basin and the Magmatic Zone: Eocene Lithospheric Stretching

Our detrital zircon U-Pb dating results demonstrate that the Yongping basin did not contain any late Cretaceous to Paleocene detritus. This indicates that the widespread late Early Cretaceous (~114 Ma) to earliest Eocene (~55 Ma) granitoids [56,60,61,87] in the regions west of the Yongping basin did not uplift and erode to provide any detritus for the Eocene deposition of this basin; that is, the Yongping basin was open to the west. Sedimentary rocks of the lower cycle do not contain any Jurassic to Early Cretaceous detritus (Figure 11A), but the middle and upper cycles do (Figure 11B,C). This indicates a provenance shift in response to a drainage expansion that evolved from a relatively small one confined to SE Tibet in the earliest stage to a large one including eastern Tibet. Regionally, the Permotriassic Jomda–Weixi–Yunxian arc developed in both eastern and SE Tibet [55], but Jurassic to Early Cretaceous volcanics and granitoids [88,89] were solely identified in eastern Tibet. The successive drainage expansion and the upwards fining and thinning of the turbidite succession suggest that the Yongping basin likely formed under an extension tectonic setting. This inference is supported by geochemical data of coeval magmatic rocks and by geophysical data.

Based on published data [59–61,87,90], bi-modal Eocene plutons (55~35 Ma) developed along the western segment of the NEE–SWW-trending magmatic zone in the regions west of the Yongping basin (Figure 1). They are gabbros and diabase dykes with $SiO_2$-content < 54% and granites with $SiO_2$-content > 70%, respectively. Most granitic plutons are A-type associated with tin mineralization [60]. Some mafic rocks with positive Nb anomaly probably originated from the asthenosphere-derived melts in an intraplate environment [59], while other gabbros were derived from an enriched lithosphere mantle [61,90].

By comparison, nearly all Eocene plutons in the eastern segment of the magmatic zone formed during the short period from 37 to 34 Ma [62,63,65]. They are high-K to shoshonitic in composition and originated from partial melting of lower crust or enriched lithosphere mantle due to asthenosphere-upwelling [56,59,63–65,91].

Deep seismic sounding along a seismic profile [92] that was deployed perpendicular to the magmatic zone revealed significantly different P-wave velocity structures in the crust beneath the SE Tibetan Plateau. A ~30 km-thick high-velocity middle and lower crust exists beneath the magmatic zone, whereas the thickness of its equivalent in the regions south of this zone is only 10–15 km (the Figure 6 of [92]). We do not know the trending of the boundary between the crusts of different P-wave velocity structures according solely to the data of this profile. However, other geophysical data may provide constraints upon the boundary. Using Rayleigh-wave phase-velocity dispersion curves, Wu et al. [22] revealed an NEE–SWW-trending lower Rayleigh-wave-velocity anomaly zone in the crust beneath the magmatic zone. Tomographic imaging by P-wave arrival time data (e.g., [19,26]) demonstrated the presence of a prominent NEE–SWW-trending zone of low wave propagation that appears at depths of 100 km in the upper mantle beneath the NEE-trending magmatic zone (the Figure 6A of [26]) (Figure 1). The spatial coincidence of the distinctive geophysical anomalies at the depth with the Eocene mantle-sourced magmatic rocks and the coeval basins at the surface suggest that the NEE–SWW-trending magmatic zone on the SE Tibetan Plateau (Figure 1) likely resulted from an NNW–SSE direction lithosphere stretching and the uprising of the asthenosphere. The eastwards younger trend of the Eocene magmatic rocks and the westward openness of the Yongping basin suggest that the lithospheric stretching and asthenosphere uprising started in the west and then migrated to the east.

### 5.3. Deformations of the Yongping Basin: Miocene Crustal Shortening and Block Rotation

Our structural analyses revealed emblematic structures of the Yongping basin with spatially variable structural patterns and strains (Figure 4). As summarized in Figure 6, the northeastern and southwestern regions of the Yongping basin exhibit dome-and-basin interference patterns. The open folds with interlimb angles of ~80° indicate a shortening of ~36%. In contrast, Domain 3 in the central region deformed by overturned folds at earlier stage, which were superposed by meter-scales reclined chevron folds. The overturned folds resulted from >80% NEE-SWW-direction shortening according to their small interlimb angles of <30°, and to the predominantly NNW–SSE trending axial planes. The sedimentary rocks with dome-and-basin structure were low-grade metamorphosed or non-metamorphosed, whereas the overturned folded horizons have been metamorphosed to a higher grade evidenced by the development of spotted gneiss (Figures 4A and 6). The spatial coincidence of the larger amount of shortening with the higher-grade metamorphism supports our suggestion that the contact-metamorphism and the deformation were simultaneous and enhanced each other. The heating and hydrothermal fluid activity due to the intrusion of the leucogranite (Figure 14A) would have metamorphosed and weakened the host sedimentary rocks where the strain was localized, leading to greater shortening (Figure 14B).

As such, the initial strain of the Yongping basin was dominated by pure shear that resulted in B-type folds of sedimentary horizons and S-type mylonite of leucogranite due to NEE–SWW compression. The later stage reclined chevron folds that superposed on the overturned folds are mostly asymmetrical with NW-vergence. This, combined with the asymmetry of the reclined curtain folds in Domain 2 (Figure 6), suggests that the later stage reclined folds formed due to right-lateral transpression. The transpression concentrated along the overturned folded regions where the mechanical nature of rocks such as anisotropy and strength had been largely changed by the earlier stage larger shortening and higher-grade metamorphism. By comparison, the open folds in the regions out of the transpressional belt were superposed by another stage open fold, giving rise to a dome-and-basin interference pattern (Figure 6). The distribution of the zoned contact

metamorphic aureoles indicates that the spotted-gneiss zone was partly destroyed by a transpressional fault along the western boundary of the intrusion (Figure 14B) as evidenced by the relatively straight western boundary in contrast to its eastern equivalent (Figure 4A).

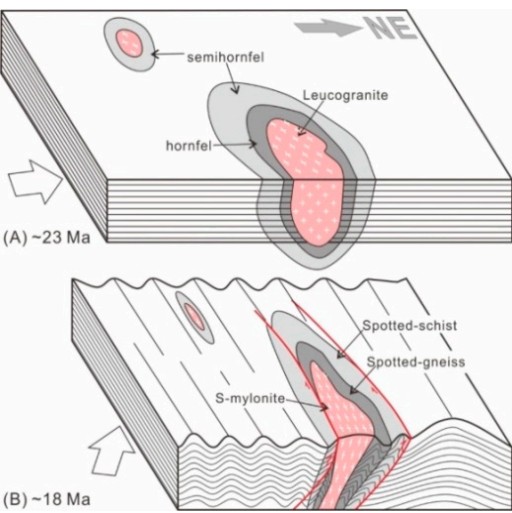

**Figure 14.** Diagrams that show the development of an antiformal "core-complex"-type structure that is common in the oblique collision belt [55] produced during syn-intrusion deformation. See text for a detailed discussion.

The above discussions demonstrate that the Yongping basin experienced a history of progressive deformation that evolved from an early stage of pure shear to a later dextral general shear. The dextral transpression occurred along a few narrow belts that likely divided the upper crust into several fragments. The fragments rotated clockwise when the narrow belts between the fragments dextrally transgressed. The published paleomagnetic data of mafic dikes with ages of ~50–35 Ma in the magmatic zone west of the Yongping basin indicate a clockwise rotation of ~87° [87], and a Middle Eocene conglomerate horizon in the central part of the magmatic zone east of the Yongping basin was rotated clockwise by ~80° [93]. Therefore, the crust of the magmatic zone was firstly fragmented and subsequently each fragment rotated with a decreasing degree to the east during the India–Eurasia collision. Li et al. [94] provided a similar scenario of orogen-scale block rotation (see their Figure 8) by a comprehensive compilation of available paleomagnetic data of SE Tibet and South China block. Such a deformation pattern of SE Tibet is consistent with its complicated seismic lithosphere structures revealed by contrast geophysical techniques [19–22].

### 5.4. Three Stages in the Dynamic Process of the India-Eurasia Collision

The above discussions demonstrate that the formation and deformation of the Yongping basin represent two sudden tectonic transitions in the evolution history of the oblique collision belt. The tectonic transitions are (1) the lithosphere started to stretch in the NNW–SSE direction at ~50 Ma, and (2) the E–W crustal shortening onset since ~23 Ma. As mentioned earlier, the orthogonal collision belt also recorded two sudden tectonic transitions at early Eocene and earliest Miocene, respectively. The sudden tectonic transitions, particularly the temporal coincidence of each transition, suggest both belts share a single time framework during their evolution, which is consistent with the known history of movement of the Indian continent [2,30–32]. Synthesizing available geological and geophysical data of both belts, the indentation of India into Eurasia can therefore be divided into the following three stages.

(1) ~60–50 Ma: Underthrust of Indian Continent Attached to Oceanic Crust

The Indian lithosphere including its continent was underthrust beneath Eurasia during the earliest stage of the India–Eurasia collision since ~60 Ma [95], and this led to eclogites [96,97] and other high-grade metamorphic rocks [98] with Indian-continental-affinity protoliths along the orthogonally collisional belt probably because of the downwards pull-force of the dense subducted oceanic crust (Figure 15A). There was neither obliquely collisional belt nor EHS before ~50 Ma.

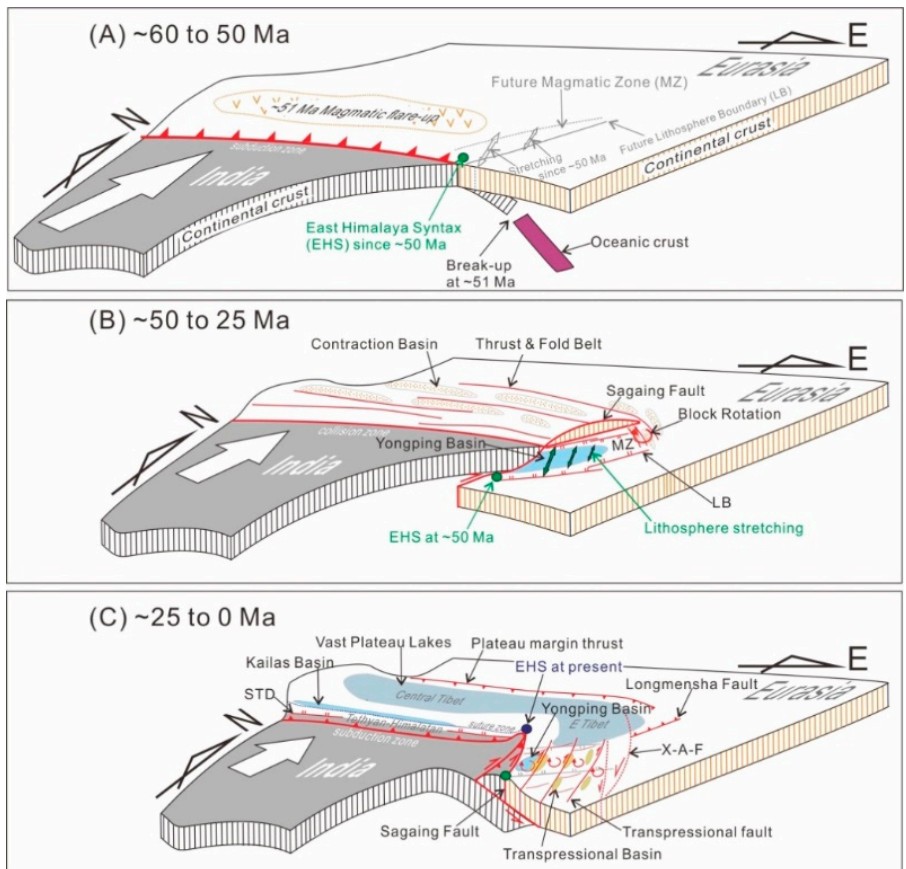

**Figure 15.** Conceptual diagrams of the three stages in the indentations of India into Eurasia since the initial contact of India with Eurasia at ~60 Ma. This model explains well the different tectonic evolution and lithospheric structures of the orthogonal and oblique collision belts in a time framework that takes into account the two tectonic transitions of both belts. See text for a detailed discussion. EHS: Eastern Himalaya syntax; LB: lithospheric boundary [17]; MZ: magmatic zone; STD: southern Tibet detachment; X-A-F: Xianshuihe-Anninghe Fault.

(2)　~50–25 Ma: Indenting of the Indian Continent as a Piston

At ~51 Ma, a magmatic flare-up developed along the orthogonal collision belt, as evidenced by a huge pile of high-K basalt, basaltic andesite, and minor dacite that is concentrated along the southern half of the Lhasa block [99] (Figure 15A). This magmatic flare-up is interpreted to be the result of the high-temperature partial melting of mantle induced by asthenosphere-sourced heating due to the break-up of the subducted oceanic slab. Sporadic occurrences of 45 Ma OIB-type gabbro [45] and coeval high Sr/Y granites in the Tethyan Himalaya [100] support this suggestion. Loss of the downwards pull-force of the denser high-pressure metamorphosed oceanic slab reduced the convergence rate by 40–45% (inset of Figure 1). Then, the Indian continent started to indent northwards into Eurasia as a piston. This is the first tectonic transition, when the Sagaing Fault, the oblique collision belt, and the EHS started to appear (Figure 15B). The compressional force of the indentation was exerted exclusively upon the orthogonal collision belt, where numerous contractional basins and associated fold-and-thrust belts developed along with the crust

shortening and thickening [35–37,67,68]. Furthermore, a torque would have appeared in the transitional belt between the orthogonal belt and the initially developed oblique collision belt. The torque progressively stretched the lithosphere in the NNW–SSE direction, which, combined with block rotation clockwise, gradually widened the transitional belt, and finally formed the NEE-SWW trending magmatic zone (Figure 15B).

(3)     ~25–0 Ma: Second Stage of Underthrust of the Indian Continent Beneath Eurasia

During the period 50 to 25 Ma, the Indian continent indented ~1500 km into Eurasia as a piston towards the NNE (14°) at a rate of ~6 cm/y [32], and this included an eastward component of ~360 km and a northward component of ~1450 km. Given a rigid indenter, the indenting would have shortened the crust of the orthogonal collision belt in the S–N direction by a maximum of ~1450 km, and of the oblique collision belt by less than 360 km in the E–W direction. The crustal shortening led to substantial crustal thickening and surface elevation predominantly along the orthogonal collision belt [39,101], because the effect of E–W direction crustal shortening (less than 360 km) of the oblique collision belt would have not balanced the effect of the NNW–SEE lithosphere stretch (up to 1450 km). Furthermore, the regions of the India–Eurasia suture zone and the Himalayan foreland were still at relatively low elevation [102]. A large elevation contrast between the Indian and Eurasia continents appeared at the earliest Miocene, thus making it possible for another stage of underthrust of the India continent to take place [48,53] (Figure 15C).

The orthogonally and obliquely collisional belts experienced different lithosphere deformation during the piston-like indenting stage (~50 to 25 Ma), resulting in the lithospheres of different structures. This may account, at least in part, for the different regimes of the second subduction of the India lithosphere beneath the different segments of Eurasia, as revealed by available geophysical data ([53] and references therein), and for their distinctive crustal deformation patterns since the earliest Miocene.

## 6. Conclusions

The crustal deformations in the orthogonal and in the oblique collision belts are distinctive because of their different tectonic locations with respect to the northward indenting India continent, but they should be inter-consistent within a single geodynamic system. In contrast to the orthogonally collisional belt where the tectonic regime remains nearly unchanged during the colliding process, the obliquely collisional belt records temporally and spatially variable deformation patterns of the lithosphere due to its northward growth. Our new model (Figure 15) provides a good interpretation of this variation. The major conclusions are summarized below.

(1)     The India–Eurasia collisional orogen composes an E–W orthogonal collision belt and an N–S oblique collision belt separated by the eastern Himalaya Syntax. Both belts record a similar collisional history of three stages separated by two tectonic transitions. However, the tectonic nature of each stage of the belts is distinctive as evidenced by their different geological and geophysical records.

(2)     The N-S oblique collision belt can be subdivided into three segments including a NEE–SWW-trending, ~250 km-wide, ~450 km-long magmatic zone in the middle. The progressive development of the magmatic zone indicates the northward growth of the oblique collision belt.

(3)     The newly identified Eocene Yongping basin is located in the central part of the magmatic zone. The Yongping basin and the magmatic zone and the oblique collision belt started to develop at ~50 Ma due to NNW–SSE lithospheric stretching.

(4)     The lithosphere of the NEE–SWW trending magmatic zone including the Eocene Yongping basin was compressed in the E–W direction since ~23 Ma. The formation and deformation of the Yongping basin represent two tectonic transitions in the evolution history of the oblique collision belt.

(5)     The process of the indentation of India into Eurasia consists of three stages. The early and late stages of lithosphere underthrust are separated by a middle stage of

continent–continent collision as a piston. Changes in the collisional dynamics between the stages correspond to the tectonic transitions.

**Supplementary Materials:** The following are available online at https://www.mdpi.com/article/10.3390/geosciences11120518/s1. (1) Supplementary Table S1. A list of published geochronological results of Cenozoic magmatic rocks, SE Tibet (including the GPS locations of samples). (2) Supplementary Table S2. U/Pb analytical results of detrital zircons from the Yongping basin, SE Tibet. (3) Supplementary Table S3. U/Pb analytical results of zircons of plutons from the Yongping basin, SE Tibet. (4) Supplementary Table S4. $^{40}$Ar/$^{39}$Ar analytical results of biotites and muscovites from the Yongping basin, SE Tibet. (5) Detailed documentation of ANALYTICAL METHODS.

**Author Contributions:** Conceptualization, T.Y. and C.X.; methodology, T.Y. and D.X.; formal analysis, D.X. and M.D.; investigation, T.Y., D.X. and C.X.; writing—original draft preparation, T.Y. and Z.Y.; writing—review and editing, T.Y., Z.Y. and D.X.; visualization, T.Y., D.X. and M.D.; project administration, T.Y. and C.X.; funding acquisition, T.Y. All authors have read and agreed to the published version of the manuscript.

**Funding:** This study is supported by the Natural Science Foundation (No. 92055206) and the Ministry of Sciences and Technology of China (No. 2016YFC0600306-4 and 2015CB452601).

**Data Availability Statement:** All the data in this study can be accessed in the supporting information appeared as the supplementary materials.

**Acknowledgments:** We are exceedingly grateful to Enrico Tavarnelli, Eduardo Garzanti Milano, Ken McCaffrey, Guillaume Duclaux, Ryan Leary, Kathryn Metcalf, and two anonymous reviewers for their very thoughtful and constructive criticism on earlier versions. Constructive comments and suggestions by three Journal's reviewers upon this paper helped to improve the manuscript.

**Conflicts of Interest:** The authors declare no conflict of interest. The funders had no role in the design of the study; in the collection, analyses, or interpretation of data; in the writing of the manuscript, or in the decision to publish the results.

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
