# Peer review of "India Indenting Eurasia: A Brief Review and New Data from the Yongping Basin on the SE Tibetan Plateau"

_geosciences, doi:10.3390/geosciences11120518_

Round 1
Reviewer 1 Report
I am glad to review this manuscript “India Indenting Eurasia: A Brief Review and New Data from the Yongping basin on SE Tibetan Plateau” by Yang Tiannan and Yanzhen et al. These authors presented great jobs on the Cenozoic basin in the southeastern Tibet. Based on their geochronological, thermochronolgical, structural and sedimentary data in the Yongping basin, they concluded that (1) this basin initially deposited at 48 Ma and was deformed at 23 Ma; (2) structural analyses of the Yongping Basin indicating E−W crustal shortening of the magmatic zone since 23 Ma. On the base of my current opinion, I find the argument logical and not difficult to follow in general. English is good. I recommend a minor correction for this manuscript based on current level.
Some comments and suggestion to revise as followed.
Line 333-334 “to spotted mica schist 333 with large biotite poikiloblasts set in a matrix of very fine biotite, muscovite, quartz, and 334 accessary minerals, displaying a diagnostic texture of contact metamorphism”, which should be given some field-based images to document. PLS add these data or other thin section observations.
Line 325-336. Domian-1 is not metamorphism at all. However, Domain-2 shows various metamorphism, such as mica schist. It is key to identify the discontinuous tectonic boundary between domain-1 and domain-2. There is a metamorphic gap between the domains! did you confirm? Please give some details about the metamorphic gap.
Figure 7. Please refine the images in Figure 7, marking the major minerals in the thin sections, some shearing senses, metamorphic minerals assemblages…. By the way, please showing your sample cites in corresponding figures.
Line 330-355. These authors should give some details on bedding orientation and structures foliation in Figure 7 in Domain-2 rocks or outcrops.
Line 348-351. “A few meter-scale intra-bedding curtain folds [80] display 348 highly variable geometries, which are limited to the thinly bedded fine-grained sand-349 stones. These folds are mostly asymmetrical, isoclinal to recumbent, and disharmonic, and 350 they commonly decrease in amplitude and fade out laterally.”. It is better to show detailed microstructures and mica-scale investigations to document these words.
Line 354. Please list the cited figure8 in order. Please check figure 8B, where it is?
Line 353 “exhibiting a tilted dome-and-basin pattern” which is a local structure in the observed cite! Pls confirm it and refined it.
Line 364 “metamorphosed channel succession”. What is “metamorphosed channel succession”?
Line 368. It should be very careful here. “Spotted gneiss” means high-temperature metamorphic condition in Domain-3. If right, it should show more details the thrust fault between domain-2 and domain-3. It is very key tectonic boundary between schist Domian-2 and amphibole phase Domain-3. The following question is “bedding S0” and structural foliation S1? These two fabrics should be investigated in details and shown here or there.
Line 387-399. Please show the leucogranite veins in Figure 6. What’s the relationships between the leucogranite veins and the S1 and S0 in Domain4 and Domain3? Please show them in field-based images and mapping job. It is strange why we can’t read the lecuogranites in Domain-3 and Domain-2? Please confirm them.
Line 394-399. Please marking the gneiss zone in crossing-section or some figures! I can’t get any information about the gneiss structure and exposed location. Pls adding them in figure 2.
About structures of leucogranites. What puzzles me is where the mylonized zone in Figure 2? These authors just presented the mylonized leucogranite bodies, in which they were hosted? What’s the relationships between the mylonite zone and the granite body and beddings? Please show all these informations in Figure 2 or crossing sections!
Line 427-432. “New grains within the quartz ribbons show highly irregular boundaries (Figs. 10B and D), suggesting that grain boundary migration (GBM) took place as the dynamic recrystallization mechanism under high-temperature conditions (500–700°C; [80]). The estimated temperature of deformation is comparable to that of the contact metamorphism, as indicated by the coexistence of andalusite and sillimanite (> 500°C) [81].” As these authors shown in Figure 10A-D, all micro-scale structures of quartz and feldspar indicate low and middle temperature (450-500 degree, can’t be beyond 500 degree) of ductile deformation corresponding to the shear in the mylonite zone. Pls confirm them. By the way, please shoing the kinematics and directions in figures.
Line 441-445, the section “brittle fracture in the beddings of the Yongping basin” can be deleted.
Line 641 “The open folds with interlimb angles of 70° indicate a shortening of36%.” Pls showing the details on the shortening estimation 36%.
Section 5.2 about “Formation of the Yongping basin…”, it is good to show some words on the formation tectonic setting of the Yongping basin, however, what’s the feature of the basin? There are other basins similar to the Yongping basin in the Western Yunan? Where they are? Why these Tertiary basins occurred along the Cenozoic large-scale shear zones, and extended parallel to the shear zone? Please discuss further.
Section 5.4.2. in the section and in Figure 16B, these authors suggested contracting setting during 50-25 Ma. Please show where the Yingping basin in this time? How to form the extensional setting to get the Yingping basin?
Author Response
Our reply immediately follows each point of reviewer's comment (we numbered each point)
Comments from Reviewer 1#
I am glad to review this manuscript “India Indenting Eurasia: A Brief Review and New Data from the Yongping basin on SE Tibetan Plateau” by Yang Tiannan and Yanzhen et al. These authors presented great jobs on the Cenozoic basin in the southeastern Tibet. Based on their geochronological, thermochronolgical, structural and sedimentary data in the Yongping basin, they concluded that (1) this basin initially deposited at 48 Ma and was deformed at 23 Ma; (2) structural analyses of the Yongping Basin indicating E−W crustal shortening of the magmatic zone since 23 Ma. On the base of my current opinion, I find the argument logical and not difficult to follow in general. English is good. I recommend a minor correction for this manuscript based on current level.
Reply: Many thanks.
Some comments and suggestion to revise as followed.
(1) Line 333-334 “to spotted mica schist 333 with large biotite poikiloblasts set in a matrix of very fine biotite, muscovite, quartz, and 334 accessary minerals, displaying a diagnostic texture of contact metamorphism”, which should be given some field-based images to document. PLS add these data or other thin section observations.
Line 325-336. Domian-1 is not metamorphism at all. However, Domain-2 shows various metamorphism, such as mica schist. It is key to identify the discontinuous tectonic boundary between domain-1 and domain-2. There is a metamorphic gap between the domains! did you confirm? Please give some details about the metamorphic gap.
Reply: Thanks for your suggestions. The first suggestion is quite consistent with our ideal that all laboratory works should be undertaken base on field relationship. In fact, the biotite poikiloblasts are generally less than 0.5 mm as shown in Figure 7A and B in our original version, and they clearly show spotted structures where micro-bedding structures are well preserved (see Fig. 7A), which are superposed by pervasive biotite poikiloblasts. Therefore, we think figures 7A and 7B are enough to show the spotted structure and it is unnecessary to add some field pictures again.
In fact, no distinct metamorphic gap between different zones of a contact metamorphic aureole was observed in the outcrops. Their metamorphic grades changed gradually, which is a major character of Barrovian type metamorphic zones. Our structural analyses revealed that the structural patterns of the domains are changed gradually as well. The displacement along the reverse fault between D-2 and D-3 is not very large. Thus we used term “reversal fault” rather than “thrust” to name this structural element.
(2) Figure 7. Please refine the images in Figure 7, marking the major minerals in the thin sections, some shearing senses, metamorphic minerals assemblages…. By the way, please showing your sample cites in corresponding figures.
Line 330-355. These authors should give some details on bedding orientation and structures foliation in Figure 7 in Domain-2 rocks or outcrops.
Reply: We sincerely think you for your remaindering.
Now we have marked the major minerals and the trace of bedding surface in these pictures in our revised version. The locations of each outcrop picture are also marked in Fig. 4. Related words in text are revised accordingly in our present version.
(3) Line 348-351. “A few meter-scale intra-bedding curtain folds [80] display highly variable geometries, which are limited to the thinly bedded fine-grained sandstones. These folds are mostly asymmetrical, isoclinal to recumbent, and disharmonic, and they commonly decrease in amplitude and fade out laterally.”. It is better to show detailed microstructures and mica-scale investigations to document these words.
Reply: Figs. 7A and B represent the microstructures of a metamorphosed pelitic sample from a tilted dome-and-basin structure (see revised Fig. 4A for their locations), but the metamorphosed sandstone does not display clear deformed structures.
(4) Line 354. Please list the cited figure 8 in order. Please check figure 8B, where it is?
(5) Line 353 “exhibiting a tilted dome-and-basin pattern” which is a local structure in the observed cite! Pls confirm it and refined it.
Reply: We made minor change of Fig. 8. Yes and it is true that this kind of structures is not pervasive. We conducted two rounds field study in 2015 and 2017, respectively. The first round field work made me quite confused. During our second round field work, the highway was being reconstructed. The previous Fig. 8C was taken in 2015 when the dome structure was quite clear. On the other hand, the previous Fig. 8B was taken in 2017 when the outer crust of the dome was damaged, but the basin structure was exposed by the highway constructing. This is the reason why these two pictures are not exactly matched.
(6) Line 364 “metamorphosed channel succession”. What is “metamorphosed channel succession”?
Reply: “channel succession” is a sedimentary term. In order to avoid any confusion, we changed this phrase to “fine clastic sediments”.
(7) Line 368. It should be very careful here. “Spotted gneiss” means high-temperature metamorphic condition in Domain-3. If right, it should show more details the thrust fault between domain-2 and domain-3. It is very key tectonic boundary between schist Domian-2 and amphibole phase Domain-3. The following question is “bedding S0” and structural foliation S1? These two fabrics should be investigated in details and shown here or there.
Reply: Yes and you are right. It is true that the gneiss formed under a higher temperature condition than the schist, but the change in temperature between them is gradual without any gap. Contact metamorphism occurs due to diffusion of heat and fluid derived from an intrusion. As such, (1) the change in metamorphic grade is gradual; and (2) there maybe not any key tectonic boundary between schist domain and gneiss domain. The bedding and schist or gneiss foliations are generally sub-parallel and are shown in Figs. 7A, 7E, and 7F.
(8) Line 387-399. Please show the leucogranite veins in Figure 6. What’s the relationships between the leucogranite veins and the S1 and S0 in Domain4 and Domain3? Please show them in field-based images and mapping job. It is strange why we can’t read the lecuogranites in Domain-3 and Domain-2? Please confirm them.
Reply: Thanks for suggestions. Figure 6 was revised following above suggestion. In fact, we tried to identify the so-called S0, S1 or S2 or S3 during our field working for structural analysis, but it is very regretful that we failed.
As shown in Fig. 7E, all metamorphic minerals are randomly oriented in a bedding surface, but they are preferred oriented in the thin-section or surface that is perpendicular to the bedding surface (e.g., Figs. 7E) defining bedding-paralleled foliation. These results demonstrate that the distribution and orientation of most metamorphic minerals were controlled by sorting of deposition rather than tectonic stress. So, it is unnecessary to differentiate S1 from S0.
(9) Line 394-399. Please marking the gneiss zone in crossing-section or some figures! I can’t get any information about the gneiss structure and exposed location. Pls adding them in figure 2.
Reply: We are sorry we cannot mark gneiss and schist units in Fig. 2 because no detailed multidiscipline job in other areas is comparable to our study.
Now we have revised Fig. 4 in our revised version, where the spotted gneiss and spotted schist as well as their spatial relation with the intrusion are displayed much clearer than the previous version.
(10) About structures of leucogranites. What puzzles me is where the mylonized zone in Figure 2? These authors just presented the mylonized leucogranite bodies, in which they were hosted? What’s the relationships between the mylonite zone and the granite body and beddings? Please show all these information in Figure 2 or crossing sections!
Reply: Thank you for your suggestions.
We all know that the result of any structural analysis strongly depends on the continuity of outcrop. The banks of Mekong River provide a relatively continuous outcrop (~ 70 %) because of highway reconstruction. Our field study results demonstrate that mylonite exclusively developed in granitic intrusion. Synthesizing structural and metamorphic data, we suggest that the mylonitization is syn-intrusion. These results demonstrate that any large-scale mylonite zone did not exist in the study area as well as in adjacent regions. Therefore, we didn’t discuss this issue in this paper due to the absence in detailed multidiscipline work along any section with continuous outcrop in other regions.
(11) Line 427-432. “New grains within the quartz ribbons show highly irregular boundaries (Figs. 10B and D), suggesting that grain boundary migration (GBM) took place as the dynamic recrystallization mechanism under high-temperature conditions (500–700°C; [80]). The estimated temperature of deformation is comparable to that of the contact metamorphism, as indicated by the coexistence of andalusite and sillimanite (> 500°C) [81].” As these authors shown in Figure 10A-D, all micro-scale structures of quartz and feldspar indicate low and middle temperature (450-500 degree, can’t be beyond 500 degree) of ductile deformation corresponding to the shear in the mylonite zone. Pls confirm them. By the way, please showing the kinematics and directions in figures.
Reply: Thanks for your suggestions.
The microstructures of quartz as shown by Figs. 10B and 10D demonstrate that the recrystallization mechanism is grain-boundary migration, which takes place under high temperature conditions according to numerous experimental results (summarized in PASSCHIER & TROUW, 2005). Generally, the internal strain of feldspar is very weak. Such phenomena are quite comparable to microstructures of some rhyolite, the latter forms due to flow of partially crystallized magma. Thus, we interpret this mylonite formed during intrusion being consistent with the occurrence of the granitic mylonite.
Now we have added arrows in Fig. 10 to show the directions of the related thin-sections in our revised version.
PASSCHIER, C. W., & TROUW, R. A. J. 2005. Microtectonics (2nd revised and enlarged edition). Springer-Verlag, Berlin, Heidelberg, New York.
(12) Line 441-445, the section “brittle fracture in the beddings of the Yongping basin” can be deleted.
Reply: Done. Now we have deleted the Fig. 11 in our previous version and rearrange the order of other figures in our present version.
(13) Line 641 “The open folds with interlimb angles of 70° indicate a shortening of 36 %.” Pls showing the details on the shortening estimation 36%.
Reply: Thanks for your suggestion.
Sorry. This is our typing errors and now we have corrected it as ‘inter-limb angles of ~80°’.
(14) Section 5.2 about “Formation of the Yongping basin…”, it is good to show some words on the formation tectonic setting of the Yongping basin, however, what’s the feature of the basin? There are other basins similar to the Yongping basin in the Western Yunnan? Where they are? Why these Tertiary basins occurred along the Cenozoic large-scale shear zones, and extended parallel to the shear zone? Please discuss further.
Reply: We sincerely thank you for your suggestions.
We are so sorry that we cannot answer these questions at present time. Just like that we have pointed out in this paper that the primary Yongping basin was likely open to the west according to our detail detrital zircon U/Pb data. This means that there probably are some remnant fragments of this basin in the regions west of our study area. As such, it likely didn’t develop along the so-called large scale shear zone. Obviously, further study is required. Basing on available geophysical and geological data, we have suggested that this basin formed due to NNW-SSE direction lithosphere stretching (see our revised version for details).
(15) Section 5.4.2. in the section and in Figure 16B, these authors suggested contracting setting during 50-25 Ma. Please show where the Yingping basin in this time? How to form the extensional setting to get the Yongping basin?
Reply: Thanks for your suggestions.
We have revised Fig. 16 in our previous version that it is now Fig. 15 in our revised version, where the location of the Yongping basin is marked.
Yes, the Yongping basin formed during the collision between India and Eurasia as a piston. Lithosphere contraction took place along the orthogonal collision belt. Meanwhile, the transitional belt between the orthogonal and oblique collision belts was stretched (extension) in NNW-SSE direction. This lithosphere stretching led to the magmatic zone as well as to the Yongping basin. For details, see our revised text.
Our model as shown by Fig. 15 in new version well explains most, if not all, available geophysical, geochemical, and geological data of both belts.
Reviewer 2 Report
I read the paper of “India Indenting Eurasia: A Brief Review and New Data from 2 the Yongping basin on SE Tibetan Plateau” by Yang et al.. Authors re-investigated the Yongping basin in the southeast of Tibet Plateau, and reviewed the subduction and collision of India plate indented into the Eurasia plate. This paper proves rich and new structural and geochronological data, which are at very high quality. The new results of field and age dating are very useful to date the deformational sequence of the Youngping basin. Data presented in the paper support conclusions in this paper, and have essential implications for re-evaluate the indentation of the India into Eurasia. Synthesizing available geological and geophysical data of both orthogonal and oblique collision belts, the indentation of India into Eurasia was therefore divided into three episodes.
This paper is good organized and written, and now is in a related good status. Figures and Tables are useful and clearly presented. Literatures were updated; however, I didn’t check all references.
I have several comments and suggestions for the further revisions of this paper, in addition, I suggest authors read and check all the spelling and writing, and polish the writing.
Comments and revision suggestions
Title: this title is ambiguous for the work of this paper and implication. I suggest reconsidering as: “Tectonics of the Yongping basin in SE Tibetan Plateau: implication for oblique collision of India into Eurasia”.
Lines 12-13, need re-writing this sentence: the contradiction between surficial and seismic lithosphere structures in the oblique collisional belt has been puzzled.
Lines 23-24, here need directly present the three stages.
Line 67, Figure 1, need minor revisions: add name of the sinistral-strike slip fault in the right corner, as well as its correlation to Red-River Fault; I could understand the meaning of Lithospheric boundary? Need more details for the LB;
Lines 102-105, Is this your three stages evolution? It’s better to name each of the stage, such as ST1, ST2 and ST3. In addition, Line 105, “remained relatively constant” is ambiguous presentation.
Line 299, Figure 4 (A), this map is a little hard to follow, because of small altitude and very complicated presentation, although authors objectively expressed geology along both banks of Mekong river. I suggest authors to present this map more concise.
Line 321, Figure 6, I suggest re-arrange the stereographic projections to make A, B, C and D correspond to Domain 1, 2, 3 and 4. Present arrangement makes reader confusion! In addition, keep consistent for the using of D-1, D-2, D-3, D-4 and Domain-1, -2, -3, and -4.
Lines 325-393, a little confusion organization for these paragraphs. I suggest authors to add principals for the division of Domain 1, 2, 3 and 4, and say the structures of Domains one by one! The present statement make reader hard to understand the differences from each one.
Lines 359 and 420, Figures 8 and 9, notice the arrangement of pictures of Figures 8 and 9.
Line 441, Are these joints or spaced cleavage in Figure 11A? pls confirming
Line 642, there are some fuzziness for the structural styles of domain 3? Pls come back to Lines 372-386 and clarify the description.
Author Response
Comments from Reviewer 2#
read the paper of “India Indenting Eurasia: A Brief Review and New Data from 2 the Yongping basin on SE Tibetan Plateau” by Yang et al.. Authors re-investigated the Yongping basin in the southeast of Tibet Plateau, and reviewed the subduction and collision of India plate indented into the Eurasia plate. This paper proves rich and new structural and geochronological data, which are at very high quality. The new results of field and age dating are very useful to date the deformational sequence of the Yongping basin. Data presented in the paper support conclusions in this paper, and have essential implications for re-evaluate the indentation of the India into Eurasia. Synthesizing available geological and geophysical data of both orthogonal and oblique collision belts, the indentation of India into Eurasia was therefore divided into three episodes.
This paper is good organized and written, and now is in a related good status. Figures and Tables are useful and clearly presented. Literatures were updated; however, I didn’t check all references.
Reply: many thanks for the reviewer’s time.
I have several comments and suggestions for the further revisions of this paper, in addition, I suggest authors read and check all the spelling and writing, and polish the writing.
Reply: Thanks. We have sent the revision to a native English speaker to polish English.
Comments and revision suggestions
(16) Title: this title is ambiguous for the work of this paper and implication. I suggest reconsidering as: “Tectonics of the Yongping basin in SE Tibetan Plateau: implication for oblique collision of India into Eurasia”.
Reply: We sincerely thank you for your good suggestions.
If we change the title of our manuscript as you suggested, it will lead to confusion in that the indenting of India into Eurasia is an oblique one. Obviously, it is not the case in fact. As having pointed out in INTRODUCTION, the indenting led to two collision belts, an orthogonal and an oblique one. Therefore, we are so sorry that we did not change the title in our revised version.
(17) Lines 12-13, need re-writing this sentence: the contradiction between surficial and seismic lithosphere structures in the oblique collisional belt has been puzzled.
Reply: Thanks for your suggestions. We rewrote ABSTRACT focusing on the two sudden tectonic transitions and our new tectonic division. Now, the above mentioned word didn’t appear in this section. In addition, we have changed “contradiction” to “inconsistence” in INTRODUCTION (line 3 to 8, Page 4 of revision).
Although having never been mentioned in any previous studies, such inconsistence suggests that our present understanding of the tectonics of the oblique collision belt is likely problematic. In contrast, available tectonic models of the orthogonal collision belt seem to be more reasonable. Our new tectonic subdivision and new data presented in this paper shed new lights on this issue. This is the reason why we don’t want to change the title as well as to accept the comments from the reviewer 3#.
(18) Lines 23-24, here need directly present the three stages.
Reply: The tectonic characters of the three stages have been stated in the sentence immediately following the mentioned sentence.
(19) Line 67, Figure 1, need minor revisions: add name of the sinistral-strike slip fault in the right corner, as well as its correlation to Red-River Fault; I could understand the meaning of Lithospheric boundary? Need more details for the LB.
Reply: Now we have marked the name of the mentioned brittle fault following your suggestion. In order to help potential readers to understand the meaning of LB, we had tried to add the lithosphere anisotropy data (not scaled) of Sol et al. (2008) across this boundary. Unfortunately, we found that the revised figure is too heavy to read. As we have pointed out in introduction, this boundary is geophysically well known although some structural geologists may have not paid enough attention to it. Factually across this boundary, other seismic structures of the lithospheres are quite distinctive. So we are sorry that we cannot follow this suggestion to do other changes in our revised version.
(20) Lines 102-105, Is this your three stages evolution? It’s better to name each of the stage, such as ST1, ST2 and ST3. In addition, Line 105, “remained relatively constant” is ambiguous presentation.
Reply: Thanks for your suggestion.
We are so sorry that we did not do it. It is because Lines 102-105 just document data rather than interpretation.
(21) Line 299, Figure 4 (A), this map is a little hard to follow, because of small altitude and very complicated presentation, although authors objectively expressed geology along both banks of Mekong river. I suggest authors to present this map more concise.
Reply: This is true and we agree with your suggestions. However, it is not very easy to get a balance between detail and conciseness.
Given the profiles (B, C, and D) and steroprojection (right-low insets), we think that the tiny symbols of altitudes maybe not a problem. We have enlarged the symbols of folds. A problem of this map concerns the presentation of the metamorphic zonings, which is not clear enough (see comment of reviewer 1#). In order to address this concern, we recompiled this map by adding some symbols to show their spatial distribution pattern.
(22) Line 321, Figure 6, I suggest re-arrange the stereographic projections to make A, B, C and D correspond to Domain 1, 2, 3 and 4. Present arrangement makes reader confusion! In addition, keep consistent for the using of D-1, D-2, D-3, D-4 and Domain-1, -2, -3, and -4.
Reply: Done. In addition, we have also added the intrusion and associated metamorphic zones in this map.
(23) Lines 325-393, a little confusion organization for these paragraphs. I suggest authors to add principals for the division of Domain 1, 2, 3 and 4, and say the structures of Domains one by one! The present statement make reader hard to understand the differences from each one.
Reply: We modified this part of text according to suggestion from a native English speaker. Besides structural patterns, we also took metamorphic mineral assemblage into account for domain division.
(24) Lines 359 and 420, Figures 8 and 9, notice the arrangement of pictures of Figures 8 and 9.
Reply: Done. See our reply to a similar comment from reviewer 1#.
(25) Line 441, Are these joints or spaced cleavage in Figure 11A? pls confirming
Reply: We have deleted the joints-related words and previous figure 11 following the suggestion from Reviewer 1#.
(26) Line 642, there are some fuzziness for the structural styles of domain 3? Pls come back to Lines 372-386 and clarify the description.
Reply: Our field work results demonstrate that the structures of Domain 3 are a little complicated, which resulted from the existence of the largest granitic intrusion. So, we have to divide this domain into two sub-domains including a western one and an eastern one (see Fig. 4). Associated text was revised following suggestion from a native English speaker.
Reviewer 3 Report
This manuscript is very interesting at it give new geochronological constraints in SE Tibet and deserves publication.
My main contest is that the title, the introduction and the discussion is too broad and merits to focus on the SE Tibet and not the overall India-Asia collision. I suggest to recentered the paper on the basin and link the structures recorded in the SE Tibet geological evolution
I will be please to review a revised and shoterned version
Author Response
comments of Reviewer 3#:
This manuscript is very interesting at it give new geochronological constraints in SE Tibet and deserves publication.
(27) My main contest is that the title, the introduction and the discussion is too broad and merits focusing on the SE Tibet and not the overall India-Asia collision. I suggest to recentered the paper on the basin and link the structures recorded in the SE Tibet geological evolution.
Reply:
As briefly mentioned in our introduction sections, the major problem of the oblique collision belt concerns the inconsistence between its surficial structures and seismic structures in depth (line 3 to 8, Page 4 of revision), whereas the both structures of the orthogonal collision belt are inter-consistent. I believe that this is a main reason why available tectonic model(s) of the orthogonal collision belt seem to be more reasonable. Although having never been mentioned in any previous studies, this inconsistence suggests that our present understanding of the tectonics of the oblique collision belt is likely problematic. According to the published structural studies, the tectonics of the oblique collision belt is featured predominantly by several largescale lateral sliding belts. Except the Sagaing fault, other lateral sliding belts likely don’t exist according to my unpublished new data. As such, it might be a large scientific risk to “link the structures recorded in SE Tibet geological evolution”.
Our most significant new findings of this paper are (1) two sudden tectonic transitions of the oblique collision belt revealed by the formation and deformation of the newly identified Yongping basin, and (2) the oblique collision belt consists of three NEE-trending fragments. These new findings cannot be interpreted in a reasonable way in the term of solely SE Tibetan geology. Given the spatial coincidence of the tectonic transitions of the oblique collision belt with their equivalents in the orthogonal collision belt, I am sorry for that we didn’t accept the suggestion.
In addition, this paper would become a local interesting one if we modify it following above suggestion.
Round 2
Reviewer 3 Report
I review for the second time this manuscript, despite few corrections, my perception of the paper is better and except one modification. I consider that this paper gives new data and deserved for publication.
I suggest to the the legend inside the figure 2